# Structures of the DarR transcription regulator reveal unique modes of second messenger and DNA binding

Maria A. Schumacher [1] ✉, Nicholas Lent[1], Vincent B. Chen [1] & Raul Salinas[1]

The mycobacterial repressor, DarR, a TetR family regulator (TFR), was the first transcription regulator shown to bind c-di-AMP. However, the molecular basis for this interaction and the mechanism involved in DNA binding by DarR remain unknown. Here we describe DarR-c-di-AMP and DarR-DNA structures and complementary biochemical assays. The DarR-c-di-AMP structure reveals a unique effector binding site for a TFR, located between DarR dimer subunits. Strikingly, we show this motif also binds cAMP. The location of the adenine nucleotide binding site between subunits suggests this interaction may facilitate dimerization and hence DNA binding. Indeed, biochemical assays show cAMP enhances DarR DNA binding. Finally, DarR-DNA structures reveal a distinct TFR DNA-binding mechanism involving two interacting dimers on the DNA. Thus, the combined data unveil a newly described second messenger binding motif and DNA binding mode for this important family of regulators.

Single-celled organisms such as bacteria are faced with constantly changing environmental conditions. Hence, they must elicit appropriate adaptive responses to survive. This is accomplished via signal transduction systems. The most common prokaryotic signal transduction mechanisms involve one-component regulatory systems that are comprised of a single polypeptide containing a sensory domain and a DNA-binding domain. Among the best studied of the single component systems are the TetR family of regulators (TFRs)[1–4]. TFRs are the third most frequently occurring transcription factor family in bacteria and regulate diverse aspects of bacterial physiology. TFRs bind and respond to a wide range of effectors, which commonly function as inducers[1–34]. Inducers are ligands that bind and inactivate repressors from interacting with target DNA operator sites. TFRs are perhaps best known for their roles in multidrug resistance (MDR), through the regulation of multidrug efflux pumps[1–10]. Recently, DarR, a 201 residue, TFR regulator of a putative MDR transporter, Ms5347, in Mycobacteria was discovered and represents the focus of this work[11].

DarR was found to not only regulate transcription of Ms5347, but also genes encoding enzymes involved in fatty acid synthesis and the cold shock protein, CspA[11]. DarR also regulates transcription of its own gene[11]. Consistent with a critical regulatory role for DarR in mycobacterial physiology, *darR* knockdown resulted in growth defects while its overexpression was toxic[11]. Interestingly, DarR was discovered in a screen in *Mycobacteria smegmatis* (now *Mycolicibacterium smegmatis*) for cyclic-di-AMP (c-di-AMP) binding factors[11]. Cyclic nucleotides, c-di-GMP and c-di-AMP, have recently emerged as key prokaryotic second messengers. While several c-di-GMP receptors have been discovered and characterized[12–16], comparatively fewer c-di-AMP binding proteins have been identified. c-di-AMP is synthesized by diadenylate cyclases via the condensation of two ATP molecules and is degraded by three families of phosphodiesterases (PDEs)[17–19]. C-di-AMP can also be removed from the cell via export by specific MDR transporters[17]. The bacterial processes regulated by c-di-AMP are diverse, ranging from the maintenance of osmotic pressure, DNA damage responses, control of central metabolism to cold shock and regulation of fatty acid biosynthesis[17–23]. More recent studies showed that c-di-AMP is essential for the survival of several Gram-positive bacteria[17–19].

The sequence of the mycobacterial c-di-AMP binding protein, DarR, places it in the TFR family of regulators[11]. TFR members contain a highly conserved structural organization[1,24–44] composed of 9 to 11 α-helices that form two domains, an N-terminal DNA-binding domain and a C-terminal inducer binding/dimerization domain[1,2,4,24–44]. The DNA-binding domain is the most conserved region amongst TFR

[1]Department of Biochemistry, Duke University School of Medicine, Durham, NC 27710, USA. ✉e-mail: Maria.schumacher@duke.edu

proteins and is composed of helices 1–3, where helices 2–3 form a helix-turn-helix (HTH) motif. The TFR C-terminal inducer binding/dimerization domain is sequentially less conserved but generally consists of a triangle-like ligand binding region composed of helices 5, 6, and 7 with the last two helices of this domain typically combining with the corresponding helices in the dimer mate to create a four-helix bundle dimerization module[1,2,4].

TFR DNA binding requires dimerization, as all TetR proteins characterized to date bind palindromic DNA, including DarR[1,2,4,11,24–44]. The DNA operator sites bound by DarR were identified by electrophoretic mobility shift assays, chromatin immunoprecipitation and DNase I protection experiments[11]. These operators, within promoters for the *darR, Ms5347* and *cspA* genes, all contain a palindromic sequence of ATACT(N)₄AGTAT (where the N indicates less conserved bases)[11]. Further studies revealed that DarR functions as a repressor[11]. Though DarR was identified as a c-di-AMP binding protein, it remains unknown how it binds this second messenger. Also unknown is how DarR recognizes its cognate DNA. To address these questions, we solved crystal structures of DarR bound to target DNA, c-di-AMP and performed complementary biochemical analyses. Our studies reveal c-di-AMP binds a newly described TFR ligand binding site in DarR. Furthermore, we show that DarR binds cAMP using the same motif and that cAMP significantly stimulates DNA binding by DarR. The DarR-DNA structure reveals a unique mode of DNA binding for a TFR protein that involves binding of two interacting DarR dimers. Thus, these studies underscore that while TFR members are among the best characterized transcription regulators, much remains to be learned about the molecular mechanisms of ligand and DNA binding by these proteins.

## Results

### Structure determination of *M. smegmatis* and *Rhodococcus* sp. USK13 DarR

To elucidate the molecular mechanisms by which DarR binds c-di-AMP and DNA we sought to obtain structures. To enhance the likelihood of obtaining crystals, we generated expression constructs for the *M. smegmatis* DarR and two other DarR orthologs. Specifically, the DarR proteins from *Mycolicibacterium baixiangningiae* and *Rhodococcus* sp. USK13, which share 84% and 82% sequence identity with *M. smegmatis* DarR, were also expressed and purified for crystallization trials. The N-terminal HTH DNA-binding regions of these proteins are conserved indicating a shared DNA binding mode (Supplementary Fig. 1). However, to analyze operator binding by these DarR proteins, we employed fluorescence polarization (FP). These studies assessed binding of the proteins to a double stranded (ds) DNA site containing the DarR operator, 5′-TAGATACTCCGGAGTATCTA-3′ (the double stranded (ds) DNA site is formed with the complementary strand)[11]. These experiments showed that all three DarR proteins bound the DNA site with essentially the same affinity, i.e. $K_d$s of $11.4 \pm 1.2$ nM, $12.4 \pm 0.7$ nM and $10.1 \pm 1$ nM for *Rhodococcus* sp. USK13 DarR, *M. baixiangningiae* DarR and *M. smegmatis* DarR, respectively (Supplementary Fig. 2; Methods).

Crystals of *M. smegmatis* DarR and *M. baixiangningiae* DarR were produced that diffracted to 3.56 Å and 1.60 Å resolution, respectively. The *M. baixiangningiae* DarR structure was solved by selenomethionine single wavelength anomalous diffraction (SAD) and refined to final $R_{work}/R_{free}$ values of 18.5%/20.2% to 1.6 Å. This structure was then used to determine the *M. smegmatis* DarR structure (Fig. 1a) (Supplementary Table 1). There are two *M. smegmatis* DarR dimers in the crystallographic asymmetric unit (ASU) and one DarR subunit in the *M. baixiangningiae* DarR structure. A *M. baixiangningiae* DarR dimer with the same assembly as the *M. smegmatis* DarR dimer is generated by crystallographic symmetry (Fig. 1a, b). Formation of the DarR dimer by interaction of the two monomers results in the burial of ~1300 Å² of surface from solvent. The structures show that, as expected, DarR belongs to the TFR family of proteins, with DALI searches revealing the TetR member showing the strongest structural homology to DarR was

the *Thermus thermophilus* HB8 PfmR protein (pdb code:3VPR). The DarR and PfmR subunits superimpose with a root mean square deviation (rmsd) of 1.9 Å for 160 corresponding Cα atoms (Supplementary Fig. 3). Similar to other TFRs, DarR has a two-domain architecture with an N-terminal, HTH containing domain and C-terminal inducer binding/dimerization domain[1,2,4]. The DarR N-terminal domain consists of helices 1-3 and the dimer domain is comprised of helices α4-α9 (Fig. 1a). In DarR, dimerization is mediated by helices α8 and α9, which interact with α8′ and α9′ (where the prime indicates the other subunit in the dimer) to form a four-helix bundle dimerization module.

### *M. baixiangningiae* DarR structure adopts inducer bound conformation

While the *M. smegmatis* and *M. baixiangningiae* DarR structures have essentially the same overall folds (Fig. 1b), the distances between their DNA-binding recognition helices differ (as measured by the distance between the two centrally located Tyr47 residues on each recognition helix). In *M. baixiangningiae* DarR the helices are separated by 50 Å while the distance between these helices of the *M. smegmatis* dimer is 41 Å (Fig. 1b). There are also conformational differences between the structures in residues 108-135. These residues are notably proximal to the ligand/inducer binding domains. However, both structures were obtained without added ligand. Analyses of the electron density in binding pockets in the *M. smegmatis* DarR structure showed water molecules. But there was no evident density for a ligand. By contrast, electron density with an unusual spirocyclic-like structure was found in the ligand-binding pocket of the *M. baixiangningiae* DarR structure near residues 108-138 (Fig. 1c; Supplementary Fig. 4). Searches revealed no metabolites from the *E. coli* expression system harboring such a structure. Based on the crystallization conditions and the chemistry of binding, the density was best fit to a complex of Tris buffer and glycerol coordinated by boron; Tris was present at a concentration of 100 mM in the crystallization solution, glycerol was in both the protein buffer and cryo-solvent and boron is a micronutrient in bacteria[45] and other organisms and is also present in glassware. Such borate complexes were previously identified in solutions of Tris, glycerol and other polyhydroxy compounds and have been structurally characterized by NMR[46].

The borate complex ligand possesses a partial negative charge (Fig. 1c, d), which is stabilized by an interaction with DarR residue Arg135 (Fig. 1c, d). In addition to Arg135, there are numerous DarR residues that contact the ligand, including Met66, His94, Asn97, His105, Val108, His109, Tyr138, Leu168 and Asn172 (Fig. 1c). Sequence alignments of DarR proteins reveals that most of these residues are conserved (Supplementary Fig. 5). The only ligand interacting residues not conserved among DarR homologs, Asn97 and Val108, have conservative substitutions that could make the same interactions. TFR proteins that regulate the transcription of MDR pump genes are known to bind substrates of their regulated pumps, which consist of a range of structurally dissimilar compounds[1,2,4]. DarR likely similarly binds a range of inducers with diverse structures that may resemble the borate complex. The substrates of the putative MDR pump regulated by DarR are currently unknown and hence future work will be needed to elucidate Ms5347 substrates and whether they function as DarR inducers.

### DarR in complex with c-di-AMP

DarR was originally identified in a UV cross-linking assay as a c-di-AMP binding protein[11]. These studies showed that DarR specifically bound c-di-AMP, as it showed no binding to c-di-GMP[11]. C-di-AMP is generated in *M. smegmatis* by the DisA c-di-AMP cyclase (WP_011731023.1). We note that both *Rhodococcus* sp. USK13 and *M. baixiangningiae* possess similar DisA proteins, WP_109326508.1 and WP_197375332.1, respectively, which each share 93% sequence identity with the *M. smegmatis* DisA (Supplementary Fig. 6). To assess c-di-AMP binding to *M. smegmatis* and *Rhodococcus* sp. USK13 DarR proteins we utilized F-c-di-AMP

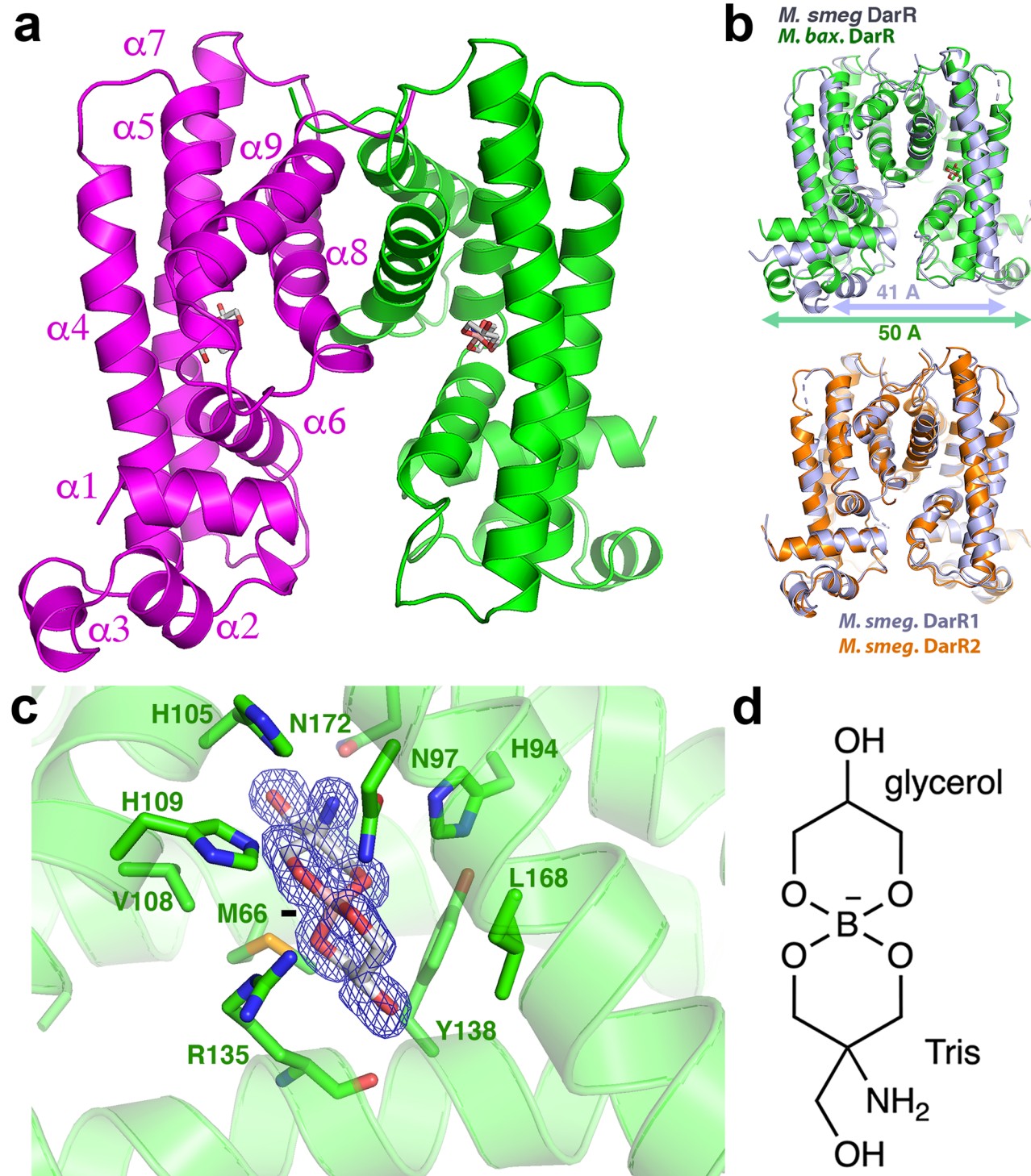

**Fig. 1 | Structures of DarR homologs. a** 1.6 Å resolution structure of the *M. baixiangningiae* DarR. One subunit is colored magenta and the other, green. Shown as sticks is the fortuitously bound ligand. Secondary structural elements are labeled for one subunit. **b** Top, superimposition of *M. baixiangningiae* DarR structure (green) onto the *M. smegmatis* structure (slate). Indicated below the structures is the distance between the recognition helices of each dimer (50 Å for the *M. baixiangningiae* DarR dimer and 41 Å for the *M. smegmatis* DarR dimer). Bottom, superimposition of the two *M. smegmatis* DarR dimers (one in slate and

one in orange) that are present in the ASU showing they are essentially identical aside from loop regions. **c** Close-up of the fortuitously bound borate complex in the *M. baixiangningiae* DarR structure with Sigma-A weighted omit electron density ($mF_o$-$DF_c$) included (blue mesh) and contoured at 3.3 σ. The omit electron density was generated in Phenix by first removing the ligand and then subjecting the coordinates to 30 cycles of refinement to remove bias. The ligand and residues that bind the ligand are shown as sticks. **d** 2-D chemical structure of the Tris-borate-glycerol complex.

(2′-O-(6-[Fluoresceinyl]aminohexylcarbamoyl)-cyclic diadenosine monophosphate) as a probe in FP studies. The proteins bound F-c-di-AMP with $K_d$s of 21.6 ± 2.5 μM and 21.9 ± 2.7 μM, respectively (Fig. 2a; Supplementary Fig. 7). The $K_d$s obtained with the fluoresceinated

probe were higher (lower affinity) than the previously 2.3 μM reported by Zheng et al. [11], which is likely due to the attached fluorescein tag. However, the probe served as a useful reporter for subsequent experiments.

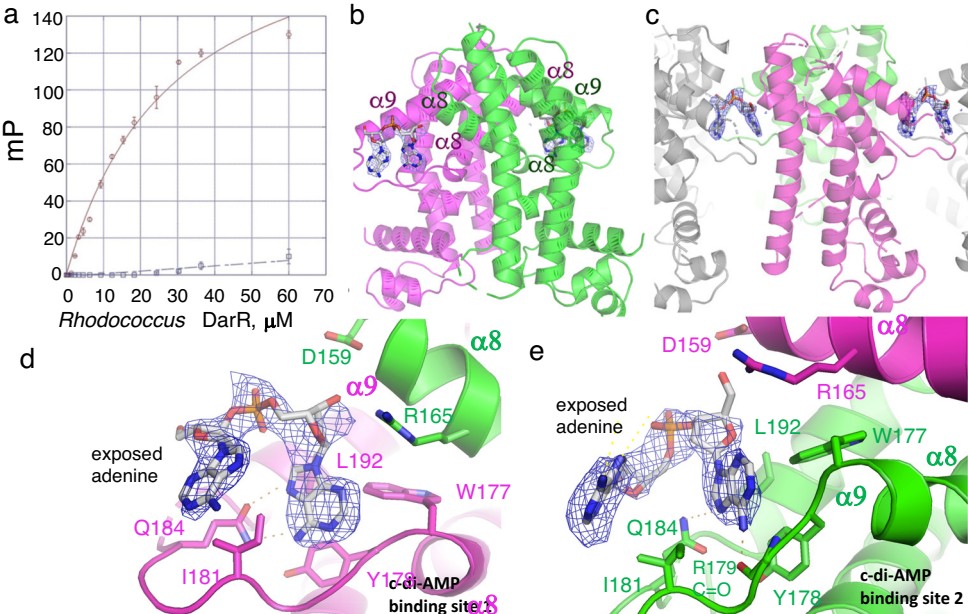

**Fig. 2 | Structure of DarR-c-di-AMP complex. a** Fluorescence polarization (FP) binding isotherms of WT *Rhodococcus* sp. USK13 DarR (red, open circles) and *Rhodococcus* sp. USK13 DarR(W177A-Q184A-L192A) (blue, open squares), respectively, to F-c-di-AMP. The x and y axes are concentration of *Rhodococcus* sp. USK13 DarR in μM and millipolarization units (mP), respectively. WT *Rhodococcus* sp. USK13 DarR bound with a $K_d$ of $21.9 \pm 2.7$ μM, while the mutant showed no detectable binding. Data points represent mean values ± SD with the error bars centered at the mean. The error in overall $K_d$ was determined as the SD between the calculated $K_d$s for three runs. **b** Overall structure of the WT *Rhodococcus* sp. USK13

DarR-c-di-AMP complex. One subunit is colored magenta and the other green. Electron density ($mF_o-DF_c$) calculated after omitting the c-di-AMP (blue mesh) is included and contoured at 2.9 σ. Helices that contain residues contributing to c-di-AMP binding are labeled. **c** c-di-AMP mediated bridging between DarR dimers observed in the crystal packing. **d** and **e** Close-up of the two nucleotide binding interactions at each of the DarR dimer interfaces with the omit electron density included (blue mesh) and contoured at 2.9 σ. One DarR subunit is colored green and the other, magenta. Residues that participate in nucleotide binding are shown as sticks and labeled.

To elucidate the structural mechanism of c-di-AMP binding we obtained the *Rhodococcus* sp. USK13 DarR-c-di-AMP structure to 2.45 Å resolution (Supplementary Table 1). The structure revealed no density for c-di-AMP in the inducer binding pocket, but density for a di-adenine nucleotide was observed near the four-helix bundle module of DarR, with the nucleotide making contacts to both subunits of the dimer (Fig. 2b–e). The density for the adenine moieties was well resolved while the density for the phosphate groups was less clear (Fig. 2b–e; Supplementary Fig. 8). Hence, a di-adenine nucleotide was fitted to the map. The phosphate moieties of the c-di-AMP appear flexible, resulting in elevated B-factors (ave) of 99.0 Å² for c-di-AMP compared to 72.2 Å² for the protein and 71.6 Å² for water molecules. In the structure, the dinucleotide is wedged within the long loop between helices α8 and α9 of one DarR subunit and also contacts residues from the N-terminus of α8´ from the other subunit (Fig. 2d–e).

In the DarR-c-di-AMP structure, a di-nucleotide is bound at each C-terminal end of the dimer (Fig. 2b, c). In this interaction, only one of the adenine bases is bound within each pocket and the other adenine base is solvent exposed (Fig. 2d–e). Each of the exposed adenine bases inserts into the binding pocket of an adjacent DarR dimer, leading to DarR polymer contacts in the crystal (Fig. 2c; Supplementary Fig. 9; Supplementary Fig. 10a, b). Comparisons of the two bound adenines in the dimer shows that the structure captured two interaction modes, in which each adenine is oriented slightly differently within the pocket. In one interaction, the adenine is specified by hydrogen bonds from the Gln184 Oε and Nε atoms to the adenine N6 and N7 atoms, respectively (Fig. 2d). In the other subunit, the N6 atom of the adenine contacts the Arg179 carbonyl oxygen and the Gln184 side chain Nε atom contacts the N7 atom (Fig. 2e). In addition to Gln184 and Arg179 both adenine moieties are contacted by the side chains of Trp177, Tyr178, Ile181 and Leu192 from one DarR subunit and Asp159 Leu162 and Arg165 from the other subunit (Supplementary Fig. 10). The Arg165 and Asp159 side

chains interact with the ribose hydroxyl group. As noted, the density for the phosphates are weak and indeed, there are no phosphate contacts from DarR.

Analyses of a multiple DarR sequence alignment shows that the residues in the loop that contact the dinucleotides are remarkably well conserved, despite being in a region that otherwise shows significant sequence variability among homologs. In particular, residues Arg165, Trp177, Gln184 and Leu192, are completely conserved (Supplementary Fig. 5). While the strong conservation of c-di-AMP binding residues lends support to our structure, to test our structural model we mutated three of the nucleotide binding residues, Trp177, Gln184 and Leu192, to alanines and performed FP binding assays with F-c-di-AMP (Fig. 2a). These experiments showed that the DarR(W177A-Q184A-L192A) triple mutant displayed essentially no binding to c-di-AMP.

## DarR binds cyclic AMP

The finding from the DarR-c-di-AMP structure that only one adenine from the dinucleotide is bound within the pocket led us to postulate that DarR might bind single adenine containing molecules. Of the adenine containing second messengers, cAMP has been shown to play a key role in mycobacterial physiology[47–61]. *M. smegmatis* encodes at least six putative adenylyl cyclases. Among these, MSMEG_3780 (AWT54739.1) has been shown to harbor adenylyl cyclase activity and to play a role in the acid stress response in *M. smegmatis*[60]. Both *Rhodococcus* sp. USK13 and *M. baixiangningiae* encode MSMEG_3780 homologs, WP213573200.1 and WP_193047576.1, respectively (Supplementary Fig. 11). We tested whether DarR could bind cAMP using the fluorescently labeled cAMP probe, 8-(2-[Fluoresceinyl]aminoethylthio)adenosine-3', 5'-cyclic monophosphate (F-cAMP) in FP studies. These experiments showed that *Rhodococcus* sp. USK13 DarR bound cAMP with a $K_d$ of $28 \pm 3$ μM (Fig. 3a).

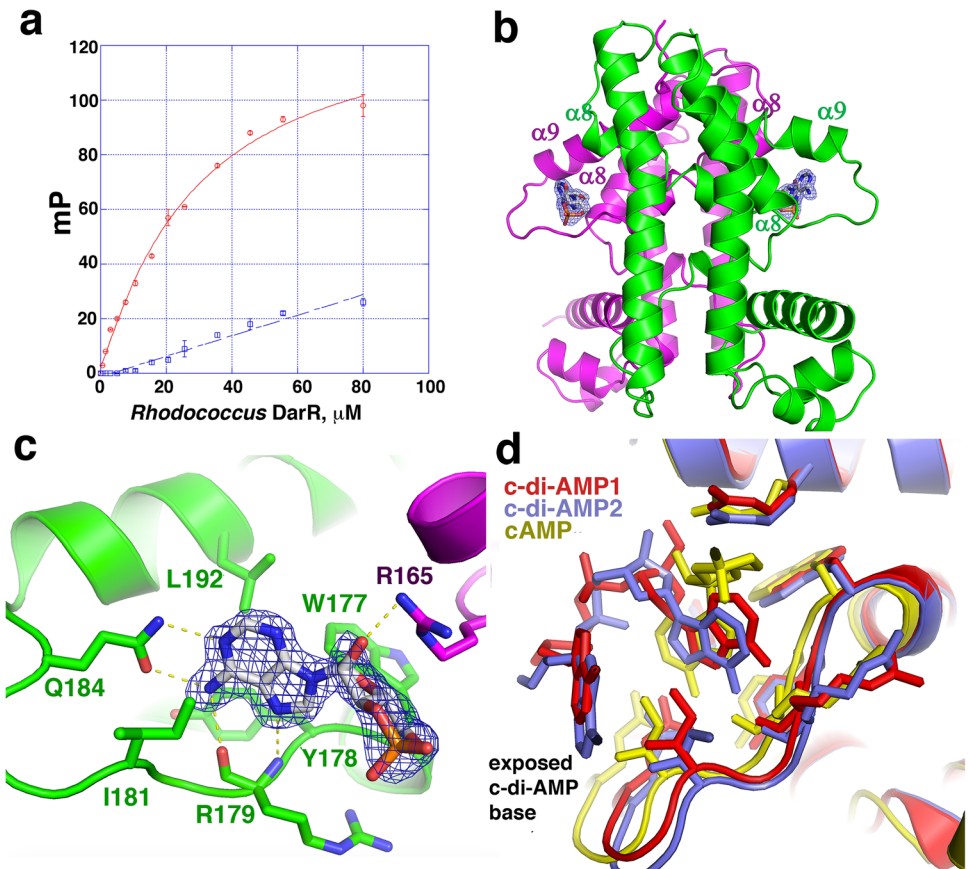

**Fig. 3 | DarR binds cAMP in the same pocket as c-di-AMP. a** FP isotherms showing binding of WT *Rhodococcus* sp. USK13 DarR (red, open circles) and *Rhodococcus* sp. USK13 DarR(W177A-Q184A-L192A) (blue, open squares), respectively, to F-cAMP. The x and y axes are concentration of *Rhodococcus* sp. USK13 DarR in μM and mP, respectively. WT *Rhodococcus* sp. USK13 DarR bound with a $K_d$ of $28 \pm 3$ μM, while the mutant showed no binding. Data points represent mean values ± SD with the error bars centered at the mean. The error in $K_d$ was determined as the SD between the calculated $K_d$s for three technical replicates. **b** Structure of the WT *Rhodococcus* sp. USK13 DarR-cAMP complex. One subunit is magenta and the other green. Sigma-A weighted omit electron density ($mF_o$-$DF_c$) is shown as a blue mesh and contoured at 3.3 σ. **c** Close up of the cAMP binding-DarR interactions with omit electron density. One DarR subunit is colored green and the other, magenta. Residues that interact with cAMP are shown as sticks and labeled. **d** Overlay of DarR-cAMP (yellow) and the two DarR-c-di-AMP bound structures (red and slate).

We next obtained the structure of the *Rhodococcus* sp. USK13 DarR-cAMP complex to 1.44 Å resolution (Fig. 3b, c; Methods). The structure revealed clear density for cAMP molecules bound at each dimer interface, in the same location bound by c-di-AMP. Unlike c-di-AMP, the cAMP appears to be tightly bound to DarR; the B-factors (ave) for the cAMP is 33.5 Å² compared to 22.6 Å² for the protein and 36.0 Å² for water molecules. Interestingly, in this structure the cAMP is rotated ~90 relative to the nucleotides in the c-di-AMP bound structure (Fig. 3d). Nonetheless, the contacts to the cAMP are provided by the same sets of residues that bind c-di-AMP. The Arg165 side chain contacts both the cAMP ribose and phosphate groups in the DarR-cAMP complex (Fig. 3b, c). The cAMP adenine N1 and N6 atoms are read by hydrogen bonds from Gln184 while the adenine N6 and N7 atoms are specified by the backbone carbonyl and amide nitrogen atoms of Arg179 (Fig. 3c). Finally, the side chains of DarR residues Trp177, Tyr178, Ile181 and Leu192 make hydrophobic interactions with the cAMP adenine moiety (Fig. 3c).

## c-di-AMP and cAMP enhance DNA binding by DarR

To our knowledge, the cyclic adenine nucleotide binding pocket we uncovered in DarR represents a new ligand binding site for a TFR protein, separate from the DNA and inducer binding sites. This new site is located between subunits within the dimerization four-helix bundle. As all characterized TFRs bind DNA as dimers we hypothesized that dimerization stabilization by nucleotide binding at this site might facilitate DNA binding. This may be particularly critical for low DarR concentrations found in vivo. Previous studies by Zhang et al., indeed, indicated that c-di-AMP binding led to enhanced interactions with DNA, however binding affinities were not determined[11]. Hence, to test our hypothesis and quantify DNA binding, we used FP binding assays and determined the $K_d$ of DarR for a 20 bp operator site in the presence of c-di-AMP and cAMP. These experiments showed that c-di-AMP and cAMP addition resulted in 2.5 and 11 fold enhancements of DNA binding; DarR bound the 20 bp operator with $K_d$s of $4.6 \pm 0.6$ nM and $1.0 \pm 0.2$ nM in the presence of c-di-AMP and cAMP, respectively, compared to $11.4 \pm 1.2$ nM in the absence of these cyclic nucleotides (Supplementary Fig. 12). Hence, cAMP significantly enhances DNA binding by DarR.

## DarR-DNA complexes reveal novel dimer-of-dimers-DNA interaction

Based on previous TFR-DNA structures, we presume that a dimeric form of DarR would bind to its operator site, which would explain the cyclic adenine mediated enhancement of DNA binding by DarR. However, to deduce the molecular mechanism of operator recognition by DarR, we next determined the structures of the *Rhodococcus* sp. USK13 and *M. baixiangningiae* DarR proteins in complex with a 20 bp site containing a double stranded (ds) DarR DNA operator, 5´-TAGA-TACTCCGGAGTATCTA-3´ (annealed to its complement). The structure of the *Rhodococcus* sp. USK13 DarR-DNA complex was solved first

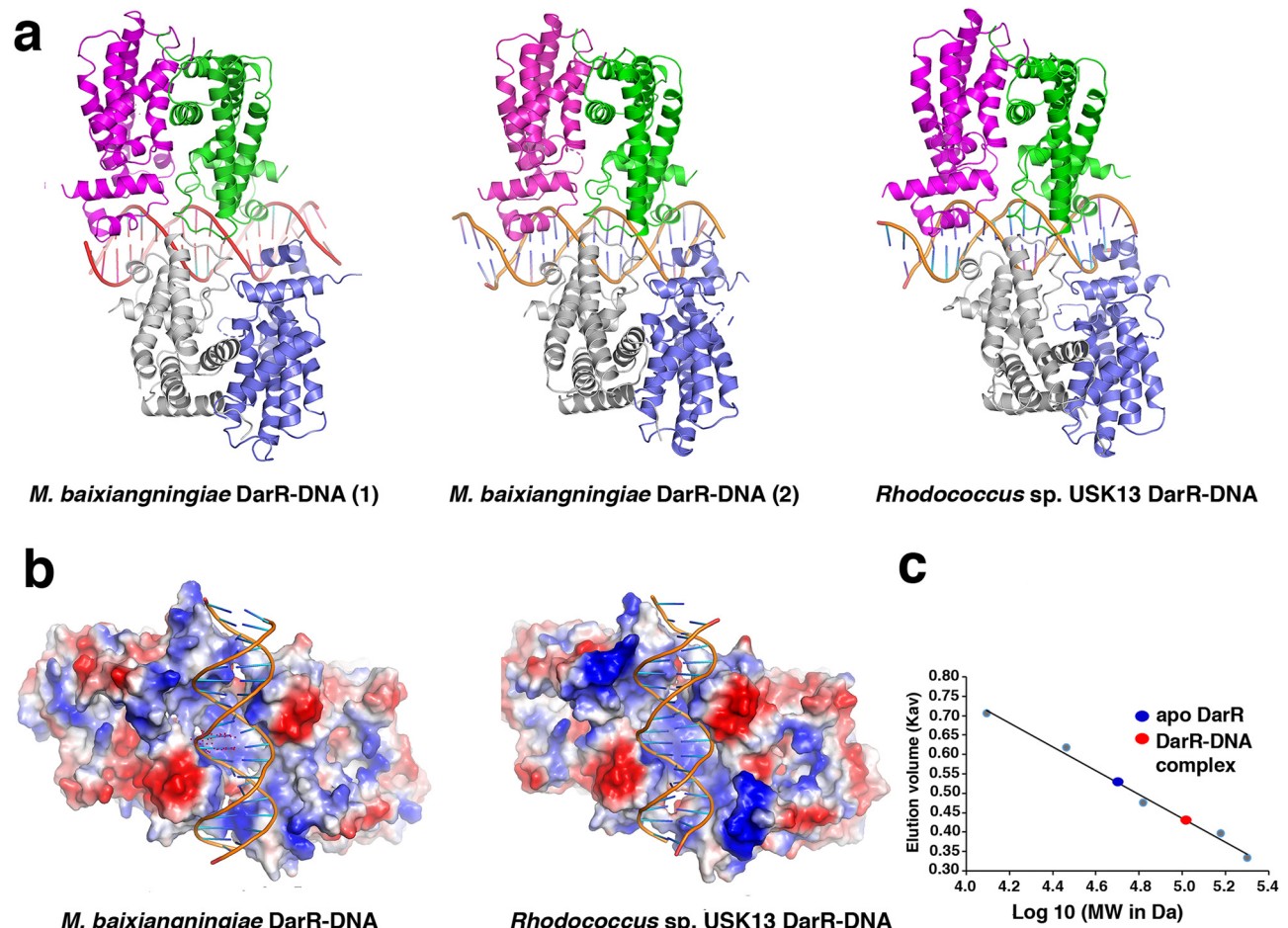

**M. baixiangningiae DarR-DNA (1)**    **M. baixiangningiae DarR-DNA (2)**    **Rhodococcus sp. USK13 DarR-DNA**

**M. baixiangningiae DarR-DNA**    **Rhodococcus sp. USK13 DarR-DNA**

**Fig. 4 | DarR-DNA structures reveal a dimer-of-dimers binding mode with cross dimer contacts. a** Ribbon diagrams of the WT *Rhodococcus* sp. USK13 DarR-DNA structure and both *M. baixiangningiae* DarR-DNA complexes present in the crystallographic asymmetric unit (ASU) showing that the complexes adopt the same dimer-of-dimer DNA binding mode. **b** Electrostatic surface representation of the WT *Rhodococcus* sp. USK13 and *M. baixiangningiae* DarR-DNA complexes rotated 90° compared to Fig. 4a. **c** Size exclusion chromatography (SEC) analyses of apo WT *Rhodococcus* sp. USK13 DarR and the WT *Rhodococcus* sp. USK13 DarR-DNA complex. The x and y axes are Log MW and elution parameter (Kav), respectively. Elution parameter Kav calculated by Kav = (elution volume for the standard − void volume)/(column volume − void volume). Apo DarR eluted (blue oval) at a calculated molecular weight (MW) of 50 kDa, consistent with a dimer, while the DarR-DNA complex (red oval) eluted at a MW of 104 kDa consistent with a DarR dimer-of-dimers-DNA complex. The standards used for calculation of the standard curve are shown (light_blue circles) and were cytochrome c oxidase (12.4 kDa), carbonic anhydrase (29 kDa), albumin (66 kDa), alcohol dehydrogenase (150 kDa) and β-amylase (200 kDa).

by selenomethionine SAD and refined to 2.96 Å resolution (Methods; Supplementary Table 2). This structure was then used to determine the 3.49 Å resolution structure of the *M. baixiangningiae* DarR-DNA complex (Fig. 4a). The *Rhodococcus* sp. USK13 DarR-DNA complex showed a pair of interacting DarR dimers in complex with the DNA (herein referred to as dimer-of-dimers) in the ASU while the *M. baixiangningiae* DarR-DNA structure contains two such complexes. These DarR-DNA complexes show the identical DNA binding mode, whereby the dimer-of-dimers partially encase the DNA using an electropositive surface (Fig. 4a, b). To assess, however, whether DarR binds DNA as a dimer-of-dimers in solution we performed size exclusion chromatography (SEC) (Fig. 4c). These analyses revealed that in the absence of DNA, the *Rhodococcus* sp. UK13 DarR eluted as a dimer (molecular weight (MW) of 50 kDa compared to the caculated value of 45 kDa), while, consistent with our structures, the DarR-DNA complex eluted at a MW consistent with a DarR dimer-of-dimers bound to DNA (104 kDa compared to the calculated value of 106.5 kDa) (Fig. 4c; Supplementary Fig. 13).

The packing in the DarR-c-di-AMP structure showed that c-di-AMP bound between dimers in the crystal leading to the formation of DarR polymers (Fig. 2c), which could possibly impact DarR repression

function. To investigate this possibility further we superimposed the polymers onto the DNA bound form of DarR (Supplementary Fig. 14). This overlay revealed that there would be clash from one polymer direction whilst the proteins in the other direction would extend from the DNA unimpeded. However, the extended polymer, due to the angle and distance from the DNA, would likely not have a significant impact on repression (Supplementary Fig. 14). By contrast, surface representation of the DarR DNA bound dimer-of-dimers shows that the DarR dimer-of-dimers almost completely engulfs an entire face of the DNA, which may facilitate its function as a repressing roadblock (Supplementary Fig. 15).

### DarR-DarR inter-subunit contacts essential for dimer-of-dimers binding mechanism

While several TFR proteins have been shown to bind DNA as dimer-of-dimers, they have not revealed significant direct contacts between dimers[27,32–36,38]. By contrast, our DarR-DNA structures reveal that the centrally bound subunits of each of DarR dimer makes critical inter-subunit contacts (Fig. 5a; 4a, b). These β-sheet like interactions are formed between residues 116-125 of each subunit (Fig. 5a). Notably, these residues are adjacent to residues that interact with the inducer

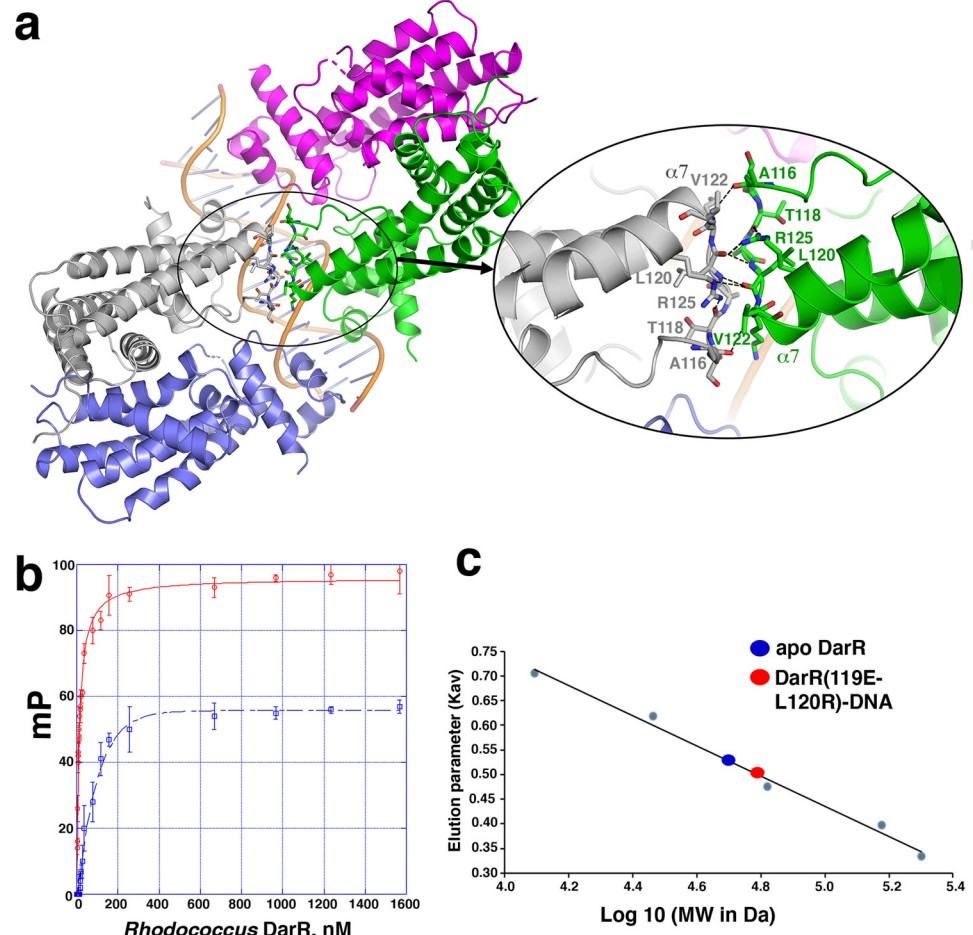

**Fig. 5 | DNA bound DarR dimers make direct protein-protein interactions.**
**a** Close-up of the interactions between the two centrally bound DarR subunits of the DarR dimer-of-dimers-DNA complex. **b** FP isotherms of WT *Rhodococcus* sp. USK13 DarR (red, open circles) and *Rhodococcus* sp. USK13 DarR(A119E-L120R) (blue, open squares), respectively, binding to the fluoresceinated 20 bp operator site. The x and y axis are concentration of *Rhodococcus* sp. USK13 DarR in nM and mP, respectively. Data points represent mean values ± SD with the error bars centered at the mean. The error in overall $K_d$ was determined as the SD between the calculated $K_d$s for three runs. The error in $K_d$ was determined as the SD between the calculated $K_d$s for three experiments. **c** SEC analyses comparing elution of WT *Rhodococcus* sp. USK13 DarR and *Rhodococcus* sp. USK13 DarR(119E-L120R)-DNA complex. The x and y axes are LogMW and elution parameter (Kav), respectively. Kav calculated by Kav = (elution volume for the standard − void volume)/(column volume − void volume). Apo DarR eluted (blue oval) at a calculated molecular weight (MW) of 50 kDa, consistent with a dimer, while the DarR(A119E-L120R)-DNA complex (red oval) eluted at a MW of 60 kDa consistent with a DarR dimer-DNA complex. The standards used for calculation of the standard curve are the same as in Fig. 4c.

ligand in the *M. baixiangningiae* DarR structure and hence would be subject to allosterism (Fig. 1a). These cross subunit contacts are mediated primarily by backbone atoms whereby the amide nitrogens of Val122 and Leu120 hydrogen bond to the carbonyl oxygens of Ala116 and Leu120 of the other subunit, respectively (Fig. 5a). Further buttressing these contacts are the side chains of Arg125, which hydrogen bond to the carbonyl oxygens of Thr118 and Leu120 (Fig. 5a).

To test the importance of the DarR cross interactions to DNA binding we generated a DarR(A119E-L120R) mutant and performed biochemical experiments. This mutant was constructed based on modeling which suggested that bulky and hydrophilic substitutions in residue 119 together with residue 120 would destabilize the interface. In particular, we hypothesized that these mutations would prevent formation of the dimer-of-dimers but should still permit binding of one DarR dimer. FP analyses showed that the DarR mutant still bound DNA, but with a 5-fold reduction in affinity (mutant $K_d = 59.5 \pm 5$ nM compared to $11.4 \pm 1.2$ nM for the WT). Notably, the final change in mP for the mutant was essentially half of the WT, suggesting a smaller protein mass was bound to the F-DNA by the mutant (Fig. 5b). This supported the hypothesis that the mutant may bind as a dimer. To test this hypothesis directly, we analyzed the DarR(A119E-L120R)-DNA

complex by SEC. These experiments showed that the DarR(A119E-L120R)-DNA complex, indeed, eluted as a dimeric-DNA complex (MW of 61 kDa compared to the calculated MW of 60 kDa for a DarR dimer bound to DNA) (Fig. 5c; Supplementary Fig. 13).

## DarR-DNA contacts

The same protein-DNA contacts are observed in the *Rhodococcus* sp. USK13 and *M. baixiangningiae* DarR-DNA structures. Hence, due to its higher resolution, we discuss DNA contacts using the *Rhodococcus* sp. USK13 structure (Fig. 6a, b). In the complex, the base interactions are all made to the major groove by residues from the recognition helix, α3, of each DarR subunit. One dimer docks onto the TAGA-**TAC**TCC◆GGA**GTA**TCTA palindrome, where bases that are specified by Gly45 and Lys44 are bold and underlined and the center of the palindrome is indicated by a diamond; Lys44 hydrogen bonds with the O6 of the guanine on the opposite strand of the C in the TXC motif, while Gly45 provides van der Waals interactions with the thymine methyl group (Fig. 6a–c). The close interaction between the Gly45 Cα atom and the thymine suggests any other residue at position 45 would prevent DNA interaction. Tyr48 and Tyr49 also make van der Waals interactions with thymine methyl groups in some of the subunits, but

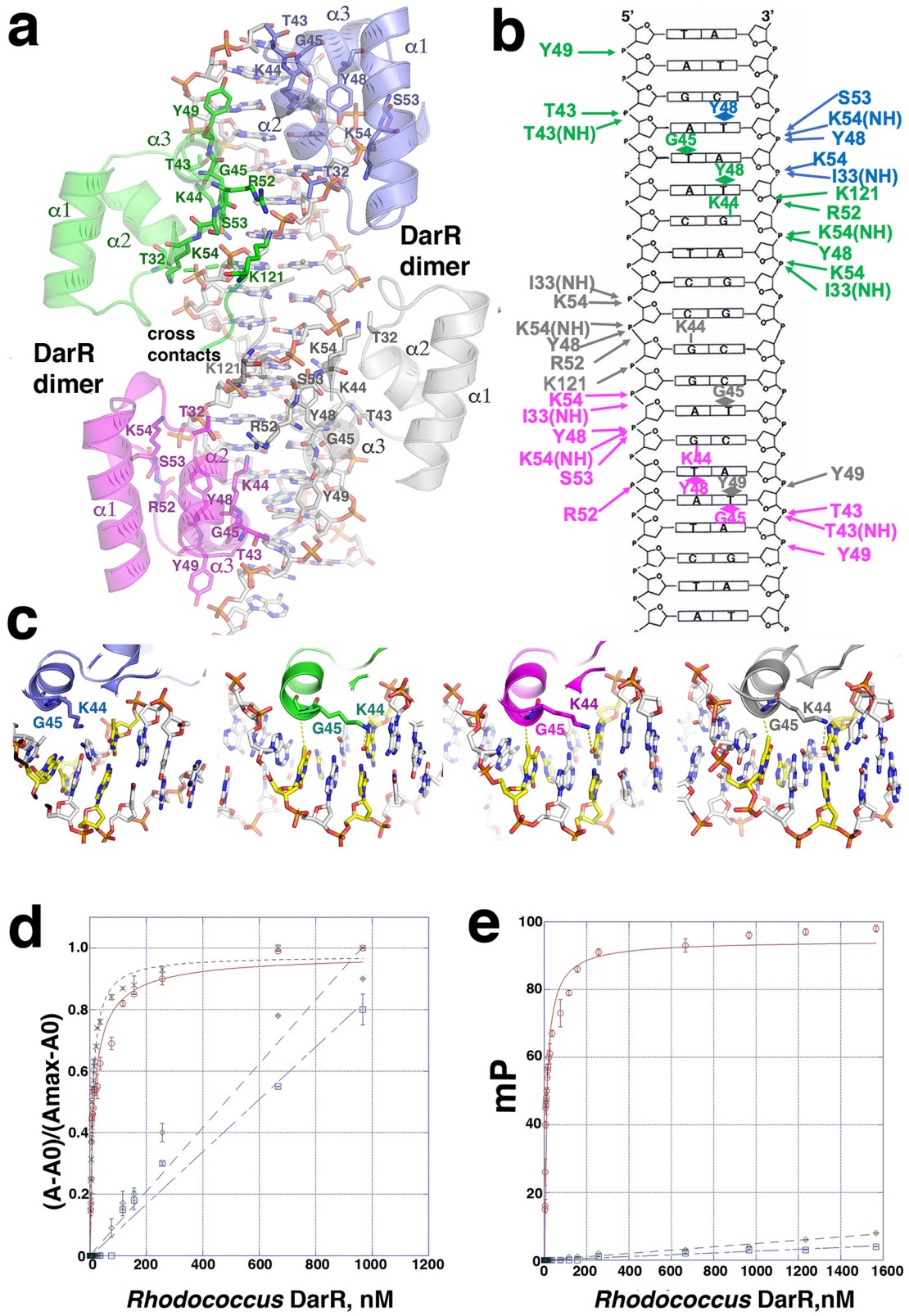

these contacts are not conserved (Fig. 6a, b). Interestingly, the second DarR dimer docks onto a DNA site, TAGATAC◆TCC**GGA**GTATCTA, that is pseudo-palindromic with the first site. Only one half site contains the TXC/GXA in this motif (underlined and bold) and the DarR subunit bound to this half site makes the same Lys44 and Gly45 contacts as the subunits bound to the first palindrome (Figs. 6c, 7a). Lys44 and Gly45 from the subunit bound to the non-palindromic half site do

not make the base contacts observed in the other half sites (colored slate in Figs. 6a–c, 7a). As a result, this subunit is weakly bound to the DNA as underscored by poor electron density for the DNA and the bound HTH (Fig. 7a).

To test the DarR-DNA structural model, we analyzed DNA binding to operator sites in which the thymines and guanines in the palindrome sites were mutated. These experiments revealed that substitutions of

**Fig. 6 | DarR-DNA contacts. a** Ribbon diagram showing DarR-DNA contacts. **b** DNA schematic showing contacts to the phosphates and bases. **c** Ribbon diagram showing the recognition helices of the DarR subunits bound to the DNA major groove and the locations and contacts of base interacting residues, Lys44 and Gly45. **d** FP isotherms for WT *Rhodococcus* sp. USK13 DarR binding to fluoresceinated 20 bp operator (top strand, 5´-TAGATACTCCGGAGTATCTA-3´) (red open circles), T mutant 20 bp operator (5´-TAG AGA CTC CGG CGT CTC TA-3´) (green open diamonds), G mutant 20 bp operator (5´- TAG ATA ATC CTG ATT ATC TA-3´) (blue open squares), optimized 20 bp operator (5´-TTG CTA CTC CGG AGT AT CTA-3´) (crosses). The *x* and *y* axes are concentration of *Rhodococcus* sp. USK13 DarR in nM and normalized D millipolarization units (mP) ((A-A$_O$)/(A$_{max}$-A$_O$)), respectively. A is change in mP reading, A$_O$ is the initial mP value before addition and A$_{max}$ is the maximal mP reading upon binding saturation. Normalization was

done here to account for slightly different A$_{max}$ values obtained for the different DNA sites. Data points represent mean values ± SD with the error bars centered at the mean. The error in K$_d$ was determined as the SD between the calculated K$_d$s for three technical replicate runs. **e** FP binding isotherms comparing the binding of WT *Rhodococcus* sp. USK13 DarR (red open circles), DarR(K44A) (green open diamonds) and DarR(G45V) (blue open squares) to the WT 20 bp operator. The x and y axes are concentration of *Rhodococcus* sp. USK13 DarR WT or mutant in nM and mP, respectively. Data points represent mean values ± SD with the error bars centered at the mean. The error in overall K$_d$ was determined as the SD between the calculated K$_d$s for three runs. The error in K$_d$ was determined as the SD between the calculated K$_d$s for the three technical replicate runs. Note, the DarR(K44A), DarR(G45V), T mutant and G mutant DNA data showed no saturable binding and hence were not fit but the points indicated with a straight line.

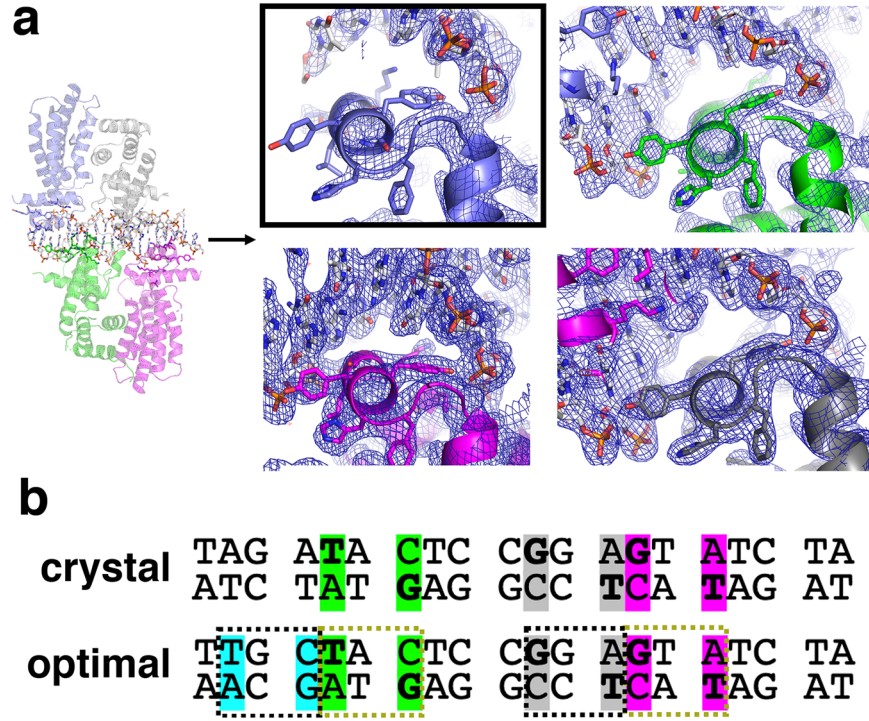

**Fig. 7 | DarR dimer-of-dimers DNA binding reveals overlapping palindromes. a** Close-up of the four *Rhodococcus* sp. USK13 DarR recognition helices docked onto the DNA. Included is a sigma-A weighted composite 2mF$_o$-DF$_c$ omit electron density map (contoured at 1 σ) for each bound subunit, which reveals that the density is weak for the DarR subunit bound to the non-optimized DNA site (outlined with a black box and colored slate). This DNA site does not contain the sequence that allows the Lys44-Guanine and Gly45-Thymine contacts observed in the other three sites. **b** Comparison of the DarR 20 bp operator site used for crystallization with the

optimized operator that showed DarR enhanced binding in Fig. 6d. Each DarR subunit binds the motif, TXC where the T is contacted by Gly45 and the G on the strand opposite the C, is contacted by Lys44. The three TXC motifs in the WT operator are indicated in green, magenta and gray. Generation of the optimal site was accomplished by adding a fourth TXC motif, which is colored blue. The yellow dashed boxes indicate the motifs contacted by subunits of the first DarR dimer while the black dashed boxes correspond to the motifs bound by subunits of the second DarR dimer.

these nucleotides prevented high affinity binding (Fig. 6d). By contrast, mutating the DNA to generate an optimized site where all four half sites contain the TXC motif led to a ~2-fold enhancement in DNA binding (K$_d$ = 6 ± 0.6 nM compared to 11 ± 1.0 nM for WT DNA) (Fig. 6d). Examination of the three characterized in vivo operator sites[11] for DarR (Supplementary Fig. 16), i.e. those from the *cspA*, *darR* and *Ms5347* promoters revealed that only the *cspA* operator site lacks a TxC half site in the second binding site (colored gray in Supplementary Fig. 16). Consistent with our structural model, EMSA studies by Zhang et al. showed that DarR bound with lower affinity to the *cpsA* operator site[11]. To further probe the structural model, we also mutated the key base specifying residues, Lys44 and Gly45 to alanine and valine residues, respectively, and showed that these mutations essentially abrogated DarR DNA binding (Fig. 6e).

In addition to base contacts, all four DarR subunits contribute phosphate contacts from the side chains of conserved residues Thr43,

Tyr48, Tyr49, Ser53, Lys54 and the amide nitrogens of Ile33, Thr43 and Lys54. Lys121 from the C-terminal region that makes cross contacts also makes phosphate interactions. This residue is either a lysine or an arginine in DarR homologs, both of which could make phosphate interactions. To analyze the DarR-bound DNA conformation for unusual features that may contribute to binding we utilized the w3DNA program[62]. These analyses showed that while the DNA is not bent and adopts an overall B-DNA conformation (rise and twist values of 3.38 Å and 33.9° compared to 3.3 Å and 34.3° for B-DNA), the AT bases in the major groove regions bound by DarR show significant propeller twist (-10° to -29°). AT-rich DNA sites are known to exhibit high degrees of propeller twist and this may play a role in allowing the interaction of these bases with DarR residues. In addition, the major grooves of the DarR-bound DNA exhibited widening compared to B-DNA. The distance between DarR recognition helices was found to be 39 Å for both DNA bound *Rhodococcus* sp. USK13 and *M. baixiangningiae* DarR

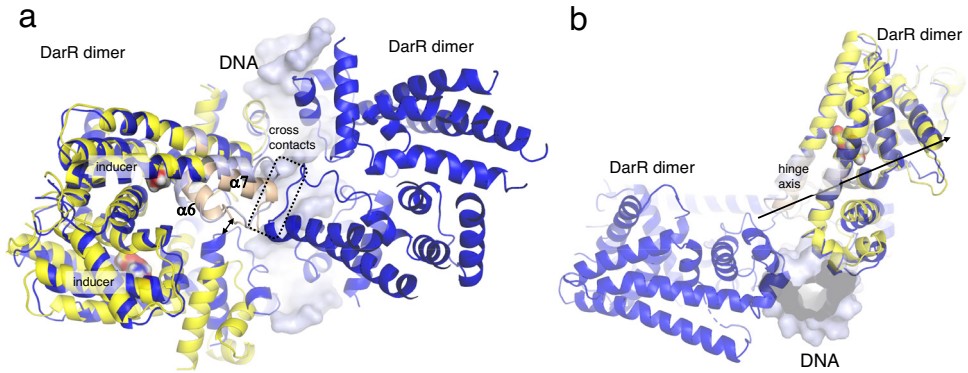

**Fig. 8 | DarR induction mechanism. a** Comparison of *M. baixiangningiae* DarR-inducer (yellow) and *M. baixiangningiae* DarR-DNA (dark blue) complexes reveal the DarR induction mechanism. The bound inducers in the DarR-inducer complex are shown as spheres. Key points of conformational change involved in the dual induction mechanism are indicated. First, the shift in the inducer bound form upon inducer binding that would eliminate the cross contacts is noted and second, a double arrow indicates the shift that leads to an increase in the distances between the HTH in the induced form. Both structural changes would abrogate DNA binding. The pivot point identified in DynDom[59], residues 108-132 are colored light_blue for the DNA-bound form and tan for the inducer bound form. **b** Side view of the complex with the molecular axis of rotation identified in DynDom shown as an arrow.

dimers. This distance is slightly larger than expected for the binding of two HTH between consecutive major grooves of B-DNA. Hence, widening of the major groove may contribute to the proper docking of DarR on operator DNA.

### DarR employs a unique dual induction mechanism

The distance between HTH motifs in the DarR-DNA complex (39 Å) is similar to that between recognition helices observed in the *M. smegmatis* DarR structure (41 Å), supporting that the latter structure adopts a DNA-bound-like state. The distance between recognition helices in the DarR-c-di-AMP and DarR-cAMP structures is ~43 Å indicating that they also adopt structures more similar to the DNA-bound state. By contrast, the HTH distance observed in the *M. baixiangningiae* DarR is significantly larger, 50 Å (Fig. 1b), indicative of an induced state. Consistent with this, the *M. baixiangningiae* DarR structure was shown to contain a bound ligand inducer (Fig. 1a–c).

This induced structure allows us to understand the induction mechanism of DarR by comparing it with the DNA bound state. These analyses revealed that ligand binding results in a large shift in residues 108-135. This leads to a relocation of the N-terminal portion of α7, residues 121-135 and a conformational change of loop residues 117-122, compared to the DNA bound state (Fig. 8a, b). These conformational changes have two consequences. First, the shift of residues in the loop towards the dimer interface leads to a pendulum-like movement of the DNA binding domain as a unit (Fig. 8a), which leads to the large increase in the distance between DNA binding domains. Second, the structural changes in residues 117-122 prevent the formation of the cross contacts between the central subunits of the dimers in the DNA bound state, which we demonstrated is essential for the dimer-of-dimer binding mechanism (Fig. 8a). The change also removes the Lys121 contact to the DNA. Hence, DarR employs a two-part induction mechanism distinct from those of other TFR proteins.

To investigate the induction mechanism in more detail we used DynDom[59] to compare the DNA- and inducer-bound states of DarR. DynDom confirmed that there is a localized hinge bending that arises when transitioning between the two states (Fig. 8b). Consistent with our analyses, DynDom[59] calculated that the rotation axis in transiting from one state to the next is composed of residues 108-132. This transition is accompanied by a rotation of 19.7° and a small translation of 0.2 Å, of the DNA-binding domains relative to the C-terminal domain. This again is a two component mechanism as in addition to repositioning of the DNA-binding domains, the movement of residues within the hinge region leads to a conformational change that prevents cross contacts.

## Discussion

TFR proteins are among the best characterized one component regulators from biological and structural standpoints. Structures of TFR proteins reveal that they share a two-domain architecture with an N-terminal DNA-binding domain and a C-terminal inducer binding/dimerization domain. Although numerous TFR structures have been determined, comparatively few have been solved in both inducer- and DNA-bound forms. Such studies are complicated by the fact that the identification of inducers is not currently predictable and hence requires experimentation. DarR, the N-terminal sequence of which places it within the TFR family, was discovered via an unusual route, which was based on its ability to bind c-di-AMP[11].

The finding that a TFR bound c-di-AMP was of interest as were data indicating that it did not appear to function as an inducer. These data suggested that DarR might bind the nucleotide using a newly described mechanism. This was confirmed by our DarR-c-di-AMP structure, which showed the second messenger does not bind within the canonical inducer binding pocket nor the DNA binding domain, but within the C-terminal four helix bundle region. Hence, the structure expands the known ligand binding/allosteric sites of TFR proteins. Studies have started to shed light on the molecular mechanisms utilized by c-di-GMP, and to a lesser extent, c-di-AMP, binding receptors. These analyses have shown that most c-di-GMP binding signatures are characterized by the presence of arginines and aspartic acids, with RxxD being the most observed c-di-GMP binding motif[12–15]. Structures of c-di-AMP bound receptors display a larger range of binding mechanisms, involving hydrogen bonds from backbone atoms and the side chains of asparagine, threonine and glutamic acid[20]. Arguably, the best-studied c-di-AMP binding motif is the regulator of conductance of K+ (RCK) domain[63–65], which is found in multiple potassium channels and transporters[63]. The RCK domain consists of an antiparallel β-sheet with a central, c-di-AMP interacting helix and c-di-AMP binds at the center of the dimer[63–65]. The RCK motif was also found to bind c-di-AMP in the recently characterized BusR transcription regulator[64]. However, the c-di-AMP binding site revealed in DarR shows no homology to any of these characterized c-di-GMP or c-di-AMP binding motifs.

Interestingly, our DarR-c-di-AMP structure showed one adenine moiety was primarily contacted by DarR with the other, solvent exposed. This led us to hypothesize and then demonstrate that DarR binds cAMP. The mid-μM binding affinity of DarR for cAMP suggests this as a functionally important second messenger as intracellular concentrations of cAMP in *M. smegmatis* have been estimated to reach

levels up to ~3 mM[60]. By contrast, c-di-AMP levels in *M. smegmatis* appear to be in the nM range[61]. Studies have shown that high target specificity for second messengers can be achieved with local signaling events between specific cyclases and target proteins[66]. However, the large difference in concentrations of the second messengers in *M. smegmatis* as well as the significant enhancement in DNA binding afforded by cAMP, would point to cAMP as a more likely physiological ligand for DarR.

As a central signaling molecule in prokaryotes as well as eukaryotes, cAMP has been the subject of intense study[47–58]. Structural analyses have revealed two common cAMP/cGMP binding motifs, the nucleotide monophosphate binding domain (CNB) and the GAF motif, which was named after the cGMP-specific phosphodiesterases, adenylyl cyclases and FhlA[57]. The CNB, which binds both cAMP and cGMP, is composed of a β-barrel surrounded by three α helices and is found in bacterial as well as eukaryotic proteins. Structures solved of CNB containing proteins include members of the bacterial CRP-FNR transcription factors and the regulatory subunit of the cAMP-dependent protein kinase in eukaryotes[47–58]. The GAF motif was only recently described and is comprised of two β-strands and α helix[57]. The cAMP binding region in DarR shows no homology to either of these motifs and instead contains a $W(X)_6Q(X)_7L$ signature and includes an arginine from the adjacent subunit. Thus, this motif consists of few conserved residues making it difficult to identify other receptors containing this signature. However, sequence analyses show that it is present in all or most DarR proteins (Supplementary Fig. 3).

Because cAMP interact with residues from both subunits of the DarR dimer, it would be predicted to function as a stabilizer of dimerization. Unfortunately, the oligomerization status of TFRs, including DarR, at the low concentrations found in cells remains largely unexplored. Indeed, solution biochemical studies, such as SEC, assessing the oligomeric states of TFRs tend to be performed at high protein concentrations that would favor dimers. Our DNA binding assays, which were carried out in the nM range, more consistent with physiological conditions, showed the cyclic adenine nucleotides enhanced DNA binding, consistent with a dimer stabilizing role. Strikingly, when we analyzed TFR structures present in the Protein Data Bank, we noticed that these proteins can be categorized into two main groups based on the different types of dimerization modules. The most abundant category, which also includes DarR, are comprised of a C-terminal region with two helices, (α8 and α9 in DarR) that combine to generate a simple four helix bundle dimerization module. A second category includes members that have extra structural elements inserted between α8 and α9 that, notably, contribute to dimerization. In particular, there are several TFR proteins that contain 2 extra helices within this region that participate in extensive dimer contacts (Supplementary Fig. 17). The binding of a small molecule, such as cAMP, that facilitates dimerization, such as in DarR, may be a way to shore up dimerization in TFR members that lack such dimer stabilizing elements.

In conclusion, our combined data have uncovered a previously unknown allosteric ligand binding site for a TFR. Our DarR-DNA and DarR-effector structures also reveal a unique dimer-of-dimers DNA binding mode and induction mechanism for a TFR protein, that involves intimate cross dimer contacts on the DNA. These combined studies thus indicate that despite the extensive structural and biological characterizations carried out on multiple TFRs, much remains to be discovered about the ligand and DNA binding mechanisms of these proteins.

## Methods

### Purification of DarR proteins

Genes encoding the *M. smegmatis* DarR, *Rhodococcus* sp. USK13 DarR and *M. baixiangningiae* DarR proteins were purchased as codon optimized genes (for *E. coli* expression) from Genscript (Piscataway, NJ, USA; http://www.genscript.com). The genes were subcloned into NdeI/

BamHI sites in pET15b, which resulted in expressed proteins with a cleavable, N-terminal Hexa-histidine tag (His-tag). *E. coli* C41(DE3) cells were transformed with these pET15b vectors. For expression of DarR proteins, cells were grown to an $OD_{600}$ of 0.6 and induced with 1.0 mM IPTG overnight at 15 °C. Cells were reconstituted into buffer A (25 mM Tris-HCl pH 7.5, 300 mM NaCl, 5% (v/v) glycerol, 0.5 mM β-mercaptoethanol (β-ME)) and lysed using a microfluidizer or sonicator. Cell debris was removed by centrifugation at 40,000 x g. The lysate, which contained the soluble DarR proteins, was loaded onto a NTA-Cobalt column and the column was washed with 500 mL of buffer A containing 15 mM imidazole. The protein was eluted in steps of increasing imidazole from 20 mM to 300 mM and fractions containing the protein were combined. At this stage the protein was >95% pure as assessed by SDS PAGE. Mutant DarR proteins were expressed and purified using the same protocol.

### DNA binding assays

To measure DNA binding to DarR, 5' fluoresceinated DNA (F-DNA) sites were used. For these experiments a buffer consisting of 25 mM N-(2-Hydroxyethyl)piperazine-N′-(2-ethanesulfonic acid) (HEPES) pH 7.5, 150 mM NaCl and 5% (v/v) glycerol was used. All the FP DNA binding experiments used F-DNA probe at a final concentration of 1 nM. For each experiment, increasing concentrations of DarR (WT or mutant) were titrated into the reaction cell. To assess the impact of c-di-AMP and cAMP on DNA binding, the adenine nucleotides were present in the buffer and protein solutions at a concentration of 1 mM. All FP experiments were conducted at 25 °C and performed in triplicate. The resultant data were plotted using KaleidaGraph to deduce binding affinities.

### c-di-AMP binding experiments

To measure c-di-AMP binding to DarR or DarR mutants, 2′-O-(6-[Fluoresceinyl]aminohexylcarbamoyl)-cyclic diadenosine monophosphate (2′-Fluo-AHC-c-di-AMP) (BioLog), was used as a fluoresceinated reporter ligand. This molecule is conjugated via a nine atom spacer to one of the c-di-AMP 2′ hydroxyl groups. The structure of the DarR-c-di-AMP complex showed that one ribose hydroxyl in the bound c-di-AMP is solvent exposed indicating it should bind DarR. These experiments were performed in a buffer consisting of 25 mM HEPES pH 7.5, 150 mM NaCl, 5% (v/v) glycerol and contained 1 nM 2′-Fluo-AHC-c-di-AMP. All FP experiments were conducted at 25 °C and performed in triplicate. The resultant data were plotted using KaleidaGraph to deduce binding affinities.

### cAMP binding experiments

To measure cAMP binding to DarR, 2′-O-(6-[Fluorosceinyl]aminohexylcarbamoyl)-cyclic diadenosine monophosphate (8-(2- [Fluoresceinyl]aminoethylthio)adenosine- 3′, 5′- cyclic monophosphate (8-[Fluo]-cAMP) (Axxora), was used as a fluoresceinated probe. This molecule contains a fluorescein attached to the C8 atom of the adenosine ring, which is solvent exposed in the structure of DarR with cAMP. FP binding experiments were carried out in a buffer consisting of 25 mM HEPES pH 7.5, 150 mM NaCl, 5% (v/v) glycerol and contained 1 nM 8-[Fluo]-cAMP. All FP experiments were conducted at 25 °C and performed in triplicate. The resultant data were plotted using KaleidaGraph to deduce binding affinities.

### Size exclusion chromatography (SEC) experiments

SEC studies were performed using a SUPERDEX™ 200 pg Hiload™ 26/600 column and an AKTA prime plus. The buffer used for the runs was 25 mM HEPES pH 7.5, 150 mM NaCl, 5% (v/v) glycerol and 0.5 mM βME. Fractions were concentrated using Sigma-Millipore concentrators (Amicon) prior to column application. Samples were loaded using a 1 mL (final volume) syringe. The SEC studies were carried out on apo *Rhodococcus* sp. USK13 DarR (at 500 μM, per monomer), the WT *Rhodococcus* sp. USK13 DarR complex with ds 20 bp DarR operator

DNA (top strand, 5´-TAGATACTCCGGAGTATCTA-3´ annealed to its complement) and DarR(L119R-A120E) bound to the 20mer. The WT DarR-DNA complex used 200 μM DarR (concentration of the monomer) to 500 μM dsDNA, the DarR(L119R-A120E)-DNA complex used 150 μM monomer protein to 350 μM dsDNA. The elution volumes of each sample were compared with a series of protein standards to determine the molecular weights. The standards used for calculation of the standard curve were cytochrome c (12.4 kDa), carbonic anhydrase (29.0 kDa), bovine serum albumin (66.0 kDa), alcohol dehydrogenase (150.0 kDa) and β-amylase (200 kDa).

### Crystallization and structure determination of apo *M. smegmatis* DarR and *M. baixiangningiae* DarR

For crystallization, the N-terminal His-tags of *M. smegmatis* DarR and *M. baixiangningiae* DarR were removed using a thrombin cleavage capture kit (EMD Millipore). The tag free proteins were then concentrated to 7 mg/mL (*M. smegmatis* DarR) and 30 mg/mL (*M. baixiangningiae* DarR) and used for screening using the hanging drop vapor diffusion method at room temperature (rt) with Wizard I-IV, Salt Rx 1 and 2, PEG Rx 1 and 2 and Natrix screens. Small crystals were identified and conditions optimized. The final crystals used for data collection of the *M. smegmatis* DarR were obtained by mixing the protein 1 to 1 with a crystallization solution of 22% (w/v) PEG 3350 and 0.1 M Citrate pH 5.6. These crystals grew within a few days and reached maximum size after 2 weeks. Optimal crystals of *M. baixiangningiae* DarR were obtained by mixing the protein 1 to 1 with a crystallization solution consisting of 2.5 M NaCl, 0.1 M Tris pH 8.5. Crystals grew after 2–5 days and continued to grow for 1 week. The *M. smegmatis* DarR and *M. baixiangningiae* DarR crystals were cryo-preserved by dipping them in a drop containing the reservoir supplemented with 20% (v/v) glycerol for 2–3 s (s) before plunging into liquid nitrogen. Data were collected at the Advanced Light Source (ALS) beamline 5.0.2 and processed with XDS (version January10, 2022)[67]. Native data sets were collected to 3.56 Å and 1.9 Å for the *M. smegmatis* DarR and *M. baixiangningiae* DarR crystals, respectively.

To solve the phase problem, selenomethionine-substituted *M. baixiangningiae* DarR was produced and used to grow crystals for single wavelength anomalous diffraction (SAD) experiments. The selenomethionine-substituted protein was expressed and purified as per the WT. After cleavage of the His-tag and concentration to 30 mg/mL, the selenomet-*M. baixiangningiae* DarR was crystallized using the same protocol as for the WT protein. The crystals were cryo-preserved as per the WT crystals and SAD data collected at ALS beamline 5.0.2 to 1.6 Å resolution. Due to the higher resolution, the selenomethionine DarR data was used in final model building and refinement. The WT structure (1.9 Å) was identical to the Semet structure. Phenix Autosol (using Phenix version 1.20.1-4487) was used to located selenium sites, perform phasing and carry out density modification[68]. The resultant experimental SAD map permitted autotracing of 80% of the structure, which contains one subunit in the asymmetric unit (ASU) in Coot (version 0.8.9.2)[69]. Crystallographic symmetry generates the DarR dimer. After correcting regions that were improperly fit and building regions that had not been autotraced, the model was further refined. At this point there was clear density in the binding pocket that had an unusual spirocyclic, ring shaped structure. After fitting individual carbon and oxygen atoms into the density, it became clear that the best fit and the one with the most chemical sense included a glycerol molecule coordinated with Tris and boron. The eLBOW software within Phenix[68] was used to generate geometry restraint information of the borate complex ligand. After inclusion of the ligand, the $R_{free}$ dropped by 0.5%. Finally, solvent molecules were added, and the structure refined to convergence.

The *M. smegmatis* structure contains two dimers in the ASU and the structure was solved using a *M. baixiangningiae* DarR subunit as the search model in Phenix[68]. Four solutions were obtained, which generated the two *M. smegmatis* dimers. After one round of refinement in Phenix, the side chains that differ in the two proteins were replaced with the *M. smegmatis* sequence and the model refined further in Phenix[68]. One dimer, in particular its DNA binding domains, had poor density, likely explaining the slightly elevated $R_{free}$. After multiple rounds of rebuilding in Coot and refinement in Phenix[68], the *M. smegmatis* DarR model converged to $R_{work}/R_{free}$ values of 27.5%/30.5% to 3.56 Å resolution. Final data collection and refinement statistics are presented in Supplementary Table 1.

### Crystallization and structure determination of *Rhodococcus* sp. USK13 DarR with c-di-AMP

For crystallization trials of *Rhodococcus* sp. USK13 DarR with c-di-AMP, the N-terminal His-tag of DarR was removed using a thrombin cleavage capture kit. For crystallization trials, the protein (at 30 mg/mL) was mixed with a final concentration of 5 mM c-di-AMP (Sigma-Aldrich; SML1231). Hanging drop vapor diffusion crystallization trials of the complex were performed at rt. Large rod like crystals were generated by mixing the complex 1 to 1 with 0.1 M Tris pH 8.5, 1.34 M lithium sulfate and 1.3% (v/v) PEG 400. Crystals grew in 2–3 days and reached maximum size in a week. The crystals were cryo-preserved by dipping them in the crystallization solution supplemented with 25% (v/v) glycerol for 2–3 s before plunging into liquid nitrogen. Data were collected at ALS beamline 5.0.1 and the structure was solved using a dimer of the *M. baixiangningiae* DarR structure as a search model. Two dimers are in the ASU and were readily placed in MR using Phenix[68]. Following an initial round of Phenix_refine[68], the side chains that differ between the two DarR proteins were replaced with those in *Rhodococcus* sp. USK13 DarR. After several rounds of refitting and refinement, clear density was observed for a di-adenine-nucleotide, which was fitted. Following water addition, the structure was refined to final $R_{work}/R_{free}$ of 21.7%/25.9% to 2.45 Å resolution (Supplementary Table 1).

### Crystallization and structure determination of *Rhodococcus* sp. USK13 DarR with cAMP

*Rhodococcus* sp. USK13 DarR in which the N-terminal His-tag was removed was used for crystallization trials with cAMP by mixing the protein (at 30-35 mg/mL) with 2 mM cAMP (Sigma-Aldrich; A6885). The mixture was used in hanging drop vapor diffusion experiments at rt. Purified *Rhodococcus* sp. USK13 DarR(K44A) with the His-tag removed was also used to generate crystals. This mutant was produced to test effects on DNA binding (the mutation is located in the DNA binding domain, far from the cAMP binding site) and was used here as a surface entropy reduction mutant[70] to assess if crystals of the mutant in complex with cAMP could be more readily obtained. Indeed, the mutant produced larger crystals of DarR in complex with cAMP, grown under the same conditions as WT, allowing for the collection of data to 1.44 Å resolution. Optimal crystals were produced using 800 mM succinic acid pH 7.0 as a crystallization solution. The crystals were cryopreserved by dipping them in a solution consisting of the crystallization reagent supplemented with 20% (v/v) glycerol for 1–2 s before plunging into liquid nitrogen. Data were collected at ALS beamline 5.0.2 and the structure was solved using a monomer of the *Rhodococcus* sp. USK13 DarR structure (from the c-di-AMP bound structure) as a search model. Crystallographic symmetry generates the DarR dimer. Following an initial round of Phenix_refine[68], density was evident for a cAMP molecule, which was fitted. After several rounds of refinement[68] and water addition, the structure was refined to convergence (Supplementary Table 1).

### Crystallization and structure determination of *Rhodococcus* sp. USK13 DarR-DNA complex

For crystallization of *Rhodococcus* sp. USK13 DarR with DNA, tag-free *Rhodococcus* sp. USK13 DarR at 30 mg/mL was mixed with 20 bp operator DNA, 5´-TAGATACTCCGGAGTATCTA-3´ (top strand annealed to its complement to generate ds blunt ended DNA) (1:1

dimer: DNA duplex) and utilized in crystallization screens using hanging drop vapor diffusion at rt. Crystals were obtained by mixing the complex 1 to 1 with solutions containing 0.1 M 2-(N-morpholino) ethanesulfonic acid (MES) pH 6.5, 30% (w/v) PEG 8000 and 0.1 M calcium acetate. Crystals grew as long rods and took 2–3 weeks to reach maximum size and were cryopreserved by dipping them in a drop of the crystallization solution supplemented with 25% (v/v) ethylene glycol. Data were collected at ALS beamline 5.0.1 and processed with XDS[67]. The structure was solved by selenomethionine SAD using data collected from a crystal grown with selenomet-substituted *Rhodococcus* sp. USK13 DarR bound to the 20 bp DNA site. Selenomethionine sites were identified and refined and density modification was performed in Phenix AutoSol[68]. While autotracing was not successful due to the low resolution, the map could be manually traced in Coot[69]. The DNA register was defined by the weaker density of the DarR subunit bound at the end of the DNA duplex with the nonoptimal site (see Results). Final data collection and refinement statistics are provided in Supplementary Table 2.

### Crystallization and structure determination of *M. baixiangningiae* DarR-DNA complex

For crystallization of *M. baixiangningiae* DarR with DNA, tag-free protein at 20 mg/mL was mixed with the 20 bp operator DNA, 5´-TAGATACTCCGGAGTATCTA-3´ (top strand annealed to its complement to generate blunt ended ds DNA) (1:1 dimer: DNA duplex). Crystallization screens were carried out using the hanging drop vapor diffusion method at rt. Crystals were obtained by mixing the complex 1:1 with solutions containing 0.1 M sodium citrate tribasic dihydrate pH 5.0, 0.2 M MgCl$_2$ and 13% (w/v) PEG 20,000. Crystals took 1–2 weeks to grow. The crystals were cryopreserved by dipping them in a drop of the crystallization solution supplemented with 25% (v/v) ethylene glycol before plunging them in liquid nitrogen. Data were collected at ALS beamline 5.0.1 and processed with XDS[67]. To solve the structure, the *Rhodococcus* sp. USK13 DarR-DNA structure was used as a search model. There are two dimer-of-dimer *M. baixiangningiae* DarR-DNA complexes in the ASU, which were successfully located in MR. After a round of refinement in Phenix_refine, the side chains were replaced for those in the *M. baixiangningiae* DarR protein. The DNA register of one complex was clearly defined by the weaker density of the DarR subunit bound at the DNA end with the nonoptimal site. The second complex was less clear and DNA register was determined by trying both possibilities with one selected based on the lower $R_{free}$ after refinement. Final data collection and refinement statistics are included in Supplementary Table 2.

### Reporting summary

Further information on research design is available in the Nature Portfolio Reporting Summary linked to this article.

## Data availability

All data generated or analyzed during this study are included in the article. Coordinates and structure factor amplitudes for the structures have been deposited in the RCSB Protein Data Bank under the accession codes 8SV6, 8SUA, 8SUK, 8T5Y, 8SVA and 8SVD. Other source data are provided as a Source_data file. Source data are provided with this paper.

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

## Acknowledgements

This research was supported by National Institutes of Health grants (R35-GM130290 and a Nanaline H Duke Professor of Biochemistry Award to M.A.S.). We acknowledge beamline 5.0.1 and 5.0.2 for X-ray diffraction data collection. The ALS (Berkeley, CA) is a national user facility operated by Lawrence Berkeley National Laboratory on behalf of the US Department of Energy under Contract DE-AC02-05CH11231, Office of Basic Energy Sciences. Beamline 5.0.2 of the ALS, a US Department of Energy Office of Science User Facility under Contract DE-AC02-05CH11231, is supported in part by the ALS-ENABLE program funded by the NIH, National Institute of General Medical Sciences, Grant P30 GM124169-01. We would like to acknowledge Dr. Kenichi Yokoyama for providing important insight into the Tris-B-glycerol ligand. We also thank Professor Richard G. Brennan for critical reading of the manuscript.

## Author contributions

M.A.S. and R.S. participated in experimental design. N.L. and R.S. made samples for structural and biochemical analyses. R.S. performed SEC experiments and performed SEC analyses. V.C. aided in determination of the Tris-B-glycerol ligand. M.A.S. performed FP analyses, grew crystals, and solved the crystal structures. M.A.S. performed analyses and wrote the paper.

## Competing interests

The authors declare no competing interests.
