## [Peer Review File · Nature Communications]

Structures of the DarR transcription regulator reveal unique modes of second messenger and DNA bindingREVIEWER COMMENTS

Reviewer #1 (Remarks to the Author):

The paper "Structures of the Mycobacterial DarR regulator reveal unique modes of second messenger and DNA binding" report crystal structures of different DarR regulators. Some of the structures are in presence of the regulator, cyclic-di-AMP, which the authors claim that this is the first structure of a transcription regulator in presence of this ligand. Similarly, structure in presence of cAMP is also presented. The overall crystallographic study has been done with sufficient rigour.

Authors obtained crystals of *M. smegmatis* DarR which diffracted upto 3.56 Å and *M. baixiangningae* DarR which diffracted much better. Authors also solved the structure of DarR from *Rhodococcus* Sp. in presence of c-di-AMP. All the structures show similar mode of binding of c-di-AMP and c-AMP to the proteins, and moreover, the modes of binding of cAMP and c-di-AMP are also similar at the same binding site.

Authors observe that c-di-AMP binds at the interface between two different dimers and propose that this leads to polymer formation. However, no data is presented to show that indeed c-di-AMP leads to aggregation of the proteins. Authors will need to demonstrate this using solution data, say by light scattering in presence of c-di-AMP, or by SEC-MALS studies.

The binding affinities of different proteins to c-di-AMP and c-AMP are not very different. This raises doubts over whether c-di-AMP is really the master regulator of these proteins, or if c-AMP can also serve the same with equal efficiency. One wonders what are the intracellular concentrations of cAMP and c-di-AMP, so that specifically only one of them would trigger transcriptional response.

From Page 12, authors consistently write that DarR binds DNA as dimer of dimers. However, it is seen that two independent dimers bind to DNA. In the larger context, the term "dimer of dimers" is typically used for tetrameric proteins, where two dimers of a protein associate with another dimer of the same protein to form a tetramer.

Dimers don't bury 1300 Å² surface from the solvent (Page 6, line 7), the two monomers in a physiological dimer do. Please correct the sentence accordingly.

There is some worry about the Apo *M. smeg* DarR data. The CC(1/2) at the cut-off resolution is 0.357, which is lower than the standards that are typically used (although there has been a debate on the inclusion of weak intensities in refinement). This is also reflected in the R_{sym} and R_{merge} values, and during refinement R_{work} and R_{free} values, which are on the higher side. Perhaps it might be good to lower the resolution limit and refine the structure against that resolution data.

In the Apo- DarR-tris-boron-glycerol structure, the α, β, γ values for trigonal space group are reported as 90, 90, 90. Authors should look into the correct assignment of space group here, or if it is simply typographical mistake.

The PDB validation report states that the space group is P3221, whereas the Table in Supplementary data mentions as P3121.

In Supplementary Figures 1 and 4, the x-axis marking intervals should be identical in all the panels.

The language of presentation needs improvement. For example, line 12 on Page 5, the sentence is terribly convoluted. Similarly, line 7 on Page 6.

Reviewer #2 (Remarks to the Author):

In their manuscript Maria Schumacher and colleagues present structures of a TetR-family transcription factor (DarR) with Ligands and supplement this structural part with some biochemical assays to address ligand and DNA binding.

DarR has been identified by Zhang et al already in 2012 as the first c-di-AMP binding transcription factor. These authors could show by crosslinking experiments that DarR binds to c-di-AMP and regulates set of genes involved in fatty acid synthesis, a cold shock protein and an MDR transporter). In more detail, c-di-AMP binding to DarR leads to an active repressor (i.e. c-di-AMP induces binding of DarR to its operator site). The manuscript by Schumacher et al. now presents structural data of DarR (in total 6 X-ray structures) of the apo-DarR protein, the DarR-cdiAMP complex and the DarR-dsDNA complex supplemented with Ligand-binding data (fluorescence polarization). The authors furthermore observe a dimer-of-dimer mechanism with dimer-DarR contacting a neighboring (second) DarR-dimer on the DNA site.

The manuscript is clearly written, and the presented structures are described in detail. The authors solved DarR structures in the different states with DarR originating from different organisms, *M. smegmatis* (Msm), *M. baixiangningiae* (Mba), *Rhodococcus* sp (Rsp), which reads a bit unusual (of course sticking to one construct/organism would be more ideal), however, if the constructs failed to crystallize this is a not too unusual approach. To justify the choice of different organisms to be part of one story, the authors provide data showing that the three orthologs behave more or less identical.

In a nutshell, the key findings are

- a) the activator site can bind an unknown ligand that remains to be identified
- b) c-di-AMP and cAMP bind to DarR at the 4helix bundle, however with medium and comparable affinities in the micromolar K_d range
- c) c-di-AMP might bridge two neighboring DarRs
- d) nucleotides slightly increase the affinity to the operator DNA
- e) dimer-of-dimer DarR on dsDNA site shows interaction of facing dimers, however the second one binds to a non-optimal sequence
- f) ligand induced distance between the HTH-sites as a regulatory model for DarR.

However, I have some points the authors should address – I have to admit it would have been easier if the pages had consecutive line numbers instead of starting with 1 on each page.

Pg2, Line 8+9: Is there something missing? The sentences read a bit truncated “TFR proteins use dimers to bind DNA”. “the binding mode suggested..”

Pg4., line 1: degraded by three families of PDEs - maybe also state that c-di-AMP can be exported by MDRs

Pg5. line 14: I know there is an alignment of many DarR homologs, but as the authors claim that *M. smegmatis*, *Rhodococcus* and *M. baixiangningiae* DarR are virtually identical, it would be extremely helpful for the reader to provide the MSA of just these three DarR proteins– or at least group them in Suppl. Fig. 3.

Maybe it is obvious, but the reader might need some info if the other species really possess a c-di-AMP pathway. I guess they all have DACs (?).

Pg5: The authors used the DarR operator site determined by Zahng for *M. smegmatis*. Are these identical for Msm and Rsp? I agree that the FP experiments show almost identical binding, but, again, it is not clear if they share the same operator site. The shape of the curves in FP looks different for Msm – but this is due to the different styles and X-axes (either use identical gridline and spacing or omit them completely, please use identical axes). Maybe I am wrong, but the fitted curve looks sigmoidal at lower concentrations (?) – which model did the authors use to fit the

data?

Is anything known about the regulated genes in *Rhodococcus* sp and *M. baixiangningiae* ? Are they identical to Msm and even more important – do they make sense in the context of c-di-AMP?

Pg6, line 9: please provide a superposition of 3VPR (supp Fig.)

Pg7: the borate complex is interesting, and it seems well coordinated in the effector binding pocket, the omit density is nice (how did the authors calculate the omit density?). It will certainly be interesting to see what the genuine ligand is. On the other hand – can the different distances between the recognition helices be explained by crystal packing?

Pg 8+9: the authors could solve the MX structure of *Rhodococcus* DarR with c-di-AMP. The density for the adenine moieties at the helix bundles seems well defined, and the fact that the loop regions become ordered upon ligand binding is a good indication for a binding site. To be able to address the ligand coordination a bit more, it would certainly be beneficial to have coordination distances in the figure (or an additional Ligand-plot). Judged from Fig. 2D the c-di-AMP is mainly kept in place by polar interactions to the sugar/phosphate and more hydrophobic environments at the base, however without the typical stacking of the aromatic residues like the W177 in the direct vicinity. Having everything of the c-di-AMP molecule in one figure is obviously tricky, but it might be hard for the reader to imagine "the truncated rest of c-di-AMP". The crystallographic table does not include it – what are the B-factors for the protein compared to c-di-AMP and waters and how does the $f_o - f_c$ -difference density look like if c-di-AMP is built in?

While I do not doubt the location of the ligand binding as the authors also test it with a mutant protein (W177A etc.), I am not really sure about the physiological role of c-di-AMP with a receptor molecule that has such an unusually low binding affinity compared to e.g. other specific receptors (*B. subtilis* DarA and also the transcription factor *S. agalactiae* BusR e.g. have low nanomolar Kds) Taking all the titrations with c-di-AMP (and also the experiments with cAMP) into account, I am not convinced that c-di-AMP is the genuine ligand but a "similar" ligand to the genuine. Being not a mycobacterium expert – what is the c-di-AMP concentrations in the cells? Do the Kds physiologically make sense in the context of the c-di-AMP concentration in the cell? In order to function as an activator for the repressor (e.g. facilitate binding to the operator site), it needs to be in the same range.

I know the c-di-AMP mediated bridging of DarR observed in the crystal-packing sounds tempting and a polymerisation behavior can indeed increase the local concentration of binding sites. However, did the authors see any oligomerisation tendency of DarR in presence of c-di-AMP in vitro to justify their polymerisation hypothesis (e.g. mass photometry, SAXS/DLS, analytical SEC(-MALLS) in presence of c-di-AMP?

The operator DNA site is also comprising just one recognition site if I got it correctly – so a tandem side-by-side DarR has no advantage over a singular one as they compete for the identical site on the same side of the DNA

Pg10. With respect to the FP-data I would assume that the Kds are in the same range. I cannot really see it in the figures, but from the figure it seems that cAMP is somewhat better coordinated and that Y178 has a stacking interaction (is it also there with c-di-AMP?). Again, a plot with ligand and coordination distances to the involved residues (and/or a ligand plot in the supplement) would be helpful.

Pg11/12: The ligand induced activation of DarR. In fact, cAMP seems to be a more potent activator of the repressor than c-di-AMP in this functional assay (also see comment to the non-saturating FP data). In think the reported Kds for the DNA-binding make sense. Did the authors test a random DNA-sequence of the same length as a control?

The crystal structures of the dimer-of-dimers are interesting, and the SEC analysis shows that DarR without DNA is dimeric and that two of these dimers can bind on one operator dsDNA. I have to admit that it is quite uncommon to perform analytical size-exclusion chromatography on a preparative 320mL 26/60 column instead of using e.g. 24mL or smaller analytical column (maybe in combination with static light scattering, because it seems that protein/DNA amount is not an

issue), but I think the Rh-derived data are ok. However, the plot Fig 4C has CV (column volumes on Y-axis) and the data presented in the supplement all show elution before 1CV (320mL column), but Fig 4C has data with 2.4 CVs in the LogMw vs CV plot – that is somehow inconsistent as DarR elutes not at 2.35 CV (=more than 650 mL). I guess there is some mix-up with the column volume.

DarR intersubunit contacts

The DarR-DarR intersubunit contacts make sense and are interesting. The mutant A119E,L120R makes sense and the FP- and SEC data show the effect of disrupting the dimer of dimers. I am not sure how this would look like in more real concentrations (the authors used 150 μ M DarR which is probably orders of magnitude beyond the concentration in the cell), but the FP data also show the effect at lower concentration.

I think suppl. Figures S8 and S7 show identical SEC-runs for the wt DarR – maybe include the mutant in S7 as S8 is mostly redundant. Even though it is only a supplement, I would definitely ask the authors for proper graphs/plots instead of providing plain screenshots taken from the chromatography software that even show (annotated) noise peaks etc.

There is one major data fitting issue in Fig 5B. The fit clearly does not describe the measured data. The measured data show a “kink” like usually seen for a stoichiometric titration (i.e. the equilibrium is always on the complex side, high affinity). It looks like the red curve (wt) shows two kinks (one at very low DarR concentration, the second one at around 150nM), while the mutant only shows one. Usually, the stoichiometry be derived from the FP data if it is a high-affinity binding (see above), but the kink is at approx. 200nM and the DNA-concentration used is 1nM? I am not sure what this means. But as long as the fit does not describe the data – it either a wrong binding model is used or something else is strange – making me believe that the Kds are different than reported.

Pg14: DarR-DNA contacts

If the second DarR only has half the contacts compared to the “first binder”, also reflected in the weak electron density, I assume that the second binding event is possible but not favoured, however stable in SEC conditions supporting that it happens at least at high concentrations. By mutation of the operator site opposite of the first binding DarR to optimise the second binding event the authors try to increase binding (“optimised” sequence). The FP data of the optimised DNA show slightly better binding compared to the genuine sequence, while the T-mutant and G-mutants disrupt binding. There are two things in the figure: the optimal sequence is not indicated by green crosses as stated in the figure legend: they are black in my PDF. And secondly, the T- and G- mutant data do not serve a fit curve as this is no binding curve. The authors should state that this line is just used for a better visualisation, but is not a binding curve.

The ligand induced change (pendulum movement) in the distance of the HTH-motifs makes sense, but also underlines that cAMP/cdiAMP are not the main inducers but rather so the ligand at the “ligand binding pocket” (*M. baixiangningiae* structure). I guess it did not crystallise and is certainly not easy to get, but the c-di-AMP+DarR+dsDNA structure would maybe also be helpful, but of course this is rather tricky, and I do not ask for it, here.

The discussion makes sense, especially with respect to the high cAMP concentrations in the cell that render the cAMP regulation of DarR more likely than being regulated by c-di-AMP. The authors clearly state that the c-di-AMP triggered polymerisation model lacks some more data at the low concentrations present in the cell (I would suggest trying mass photometry).

Figures in general:

The figures, especially the layout of graphs would benefit from some corrections. If the authors like the gridlines that is fine (but not necessary), but they should have uniform spacing and X-axes (especially if they show comparable experiments like titrations). I also assume that the “micro” (i.e. the Greek character μ) is available in any plotting/fitting software package – the authors should use of μ M instead of uM

Reviewer #3 (Remarks to the Author):

DarR was identified in *Mycobacterium smegmatis* as a cyclic-di-AMP binding protein, the first receptor found for this bacterial signalling molecule [citation #11, 2013]. Here, the authors describe six crystal structures of DarR, a TetR family regulator, from three bacterial species (*M. smegmatis*, *M. baixiangningiae*, and *Rhodococcus* sp. USK13, a Gram-positive bacterium related to mycobacteria), together with data from DNA- and nucleotide-binding experiments and size-exclusion chromatography.

The key results are:

- (1) The "apo" structures of the Msm and Mba proteins show that DarR forms a dimer, as expected for TetR family members, but the positions of the DNA binding domains of the two proteins differ by $\sim 10\text{\AA}$.
- (2) The "apo" structure of Mba-DarR shows density in the expected inducer-binding pocket and has been interpreted as a complex of tris buffer and glycerol bound to boron.
- (3) Two structures of the Rho-DarR protein crystallized with cyclic-di-AMP and cAMP show density at a site that is different from the expected inducer-binding pocket.
- (4) Cyclic-di-AMP and cAMP enhance the binding affinity of Rho-DarR to DNA containing the operator sequence by 3- and 11-fold, respectively, from a K_d of 11 nM in the absence of nucleotide.
- (5) Crystal structures of Mba- and Rho-DarR bound to DNA show two dimers of DarR binding to each DNA fragment. The structures show a protein-protein contact between one monomer of each dimer.
- (6) Size-exclusion chromatography indicates that DarR is a dimer in the absence of DNA but is a tetramer (dimer-of-dimers) when bound to DNA.
- (7) Rho-DarR containing mutations designed to disrupt the interaction between dimers binds DNA with a 6-fold lower affinity, and size-exclusion chromatography is consistent with a single dimer of the mutated DarR binding to DNA.
- (8) In the DNA co-crystals structures, one DarR dimer binds to the palindromic consensus binding site while the other dimer binds to a pseudo-palindromic sequence that is shifted relative to the other site by 3 basepairs. DarR binds ~ 2 -fold more tightly to DNA mutated to optimize the second binding site.

Based on the distances between the DNA binding domains in each structure, the authors conclude that the apo Msm-DarR structure is in a DNA-bound conformation, with a 40\AA distance between the helix-turn-helix motifs; the two Rho-DarR structures with bound nucleotides are in similar conformations, with $\sim 43\text{\AA}$ distance between the HTH motifs. In the "apo" Mba-DarR structure (with the tris-glycerol-boron complex fortuitously bound to the canonical TetR inducer-binding pocket), the HTH motifs are $\sim 50\text{\AA}$ apart, compared to the $\sim 40\text{\AA}$ distance in the Mba- and Rho-DarR complexes with DNA. The authors propose that apo-DarR binds to DNA, repressing transcription and that cyclic-di-AMP or cAMP stabilize this binding. Binding of an inducer, which has not been identified, would cause the type of conformational change that is observed in the "apo" Mba-DarR structure, releasing DarR from DNA.

Overall, the structural and biochemical data presented are sound and present an intriguing model for the regulation of DarR. The structures have been determined at resolutions ranging from 1.44 to 3.56 \AA , with the apo *M. smegmatis* structure being the least well determined. However, the biological/physiological significance of these findings is unclear for the following reasons:

- DarR homologs are not found in the pathogenic mycobacteria, including *M. tuberculosis* and *M. leprae*.
- It is unclear if cyclic-di-AMP is the nucleotide bound to the *Rhodococcus* DarR complex. On page 8, lines 21-22, the authors say that the density for the adenines was well resolved, but "the density for parts of the phosphate groups was less clear (Fig. 2B-E)." From the images provided, it appears that the dinucleotide is not cyclic. The source of the c-di-AMP used for crystallization is not

given in the methods section.

- The binding of two DarR dimers to each DNA seems serendipitous. Examples of naturally occurring sequences that contain the kind of overlapping operator sites that are present in the DNA sequence used for the crystallization experiments would significantly increase the relevance of this binding mode.

- As the authors acknowledge, it is unclear how the concentrations of nucleotides used in these experiments (2-5 mM during crystallization) relate to those that are likely to be present in mycobacterial cells.

Additional comments and questions:

1. Some of the references cited appear to be incorrect for what the author's claim they are citing. For example, in the discussion (page 18, lines 15-17), this sentence cites references 61 and 62:

"The mid- μ M binding affinity of DarR for cAMP suggests this may be a functionally important interaction as intracellular concentrations of cAMP in *S. smegmatis* have been estimated to be as high as 100 μ M."

However, neither reference ever makes the claim of cAMP concentrations in *M. smegmatis*. Reference 61 is specific to *M. bovis* BCG and *M. tuberculosis* cAMP levels, while reference 62 is about cytokinin signaling and does not mention the words "cAMP" or "cyclic" in the paper. All citations should be checked.

Note: "*S. smegmatis*" in the above sentence should be corrected to "*M. smegmatis*".

2. Since DarR is not ubiquitous among Mycobacterial species and the bulk of the structures presented in this paper are of Rhodococcus DarR, the title seems a bit misleading.

3. Is there evidence of any cooperative DNA binding by DarR, especially in the presence of c-di-AMP or cAMP? Do any other adenosine compounds bind to DarR, such as AMP or ATP?

4. Page 8, lines 2-4: Ms5347 is annotated as an MDR pump. If there are not experimental data testing this, it should be referred to as a putative pump.

5. Page 8, line 13: Include the K_d that was previously reported so that the reader does not have to look it up.

6. Page 9, lines 2-3: How is it known that the loops in the Msm and Mba DarR proteins becomes ordered upon cyclic-di-AMP binding when only the structure of the Rho-DarR protein has been determined with nucleotide bound?

7. Page 9, lines 5-21: This paragraph is rather confusing. One adenine of the di-AMP is described as being bound in a pocket and the other as being solvent exposed, but later in the paragraph the solvent-exposed adenosine is described as being inserted into the binding pocket of an adjacent DarR dimer. How can it be both solvent exposed and inserted into the binding pocket of a symmetry-related molecule? Is this binding mode possible in the dimer of dimers bound to DNA that is described later? Given the view of the di-adenine shown in Fig. 2 D-E, it does not seem possible for it to be cyclic. The figures should show the contacts between the nucleotide and the symmetry mate as well as the contacts that are shown. What part of the protein could the second phosphate be contacting, if it is still present and the di-nucleotide is still cyclic? It is not clear how far apart this binding site is from the canonical ligand binding site. Please clarify.

8. Page 14, lines 9-23, and page 15, lines 1-11: This paragraph is also confusing. The DNA sequence shown on line 13 does not match the sequence shown in Fig. 6B. It appears to be the "optimized" sequence that is shown in Fig. 7B. Since the mutated sequences shown are not fully self-complementary, the manuscript should clarify that two complementary DNA oligos are

annealed together to form a fully base paired duplex (like what is shown in Fig. 7B for the optimized sequence), if that is what was done. Adding the duplex sequences of the mutants to the figure would be helpful. Fig. 6 A and B should more clearly show which DarR molecules make up the dimer, since the dimerization domains are not visible. As labeled, Fig. 6A suggests that the "cross contacts" are between the green and magenta monomers, but aren't those the monomers that make up one dimer? Shouldn't there also be a symmetric set of cross contacts? The density shown in Fig. 7A would seem better to include in a figure with panels 6A, B and C.

9. Fig. 8 is not as informative as it could be. Where are the ligand binding sites relative to the structures shown? Adding a DynDom analysis of the movement would be informative.

We would like to thank all the reviewers for their excellent comments and suggestions, which have led to an improved manuscript.

REVIEWER COMMENTS

Reviewer #1 (Remarks to the Author):

The paper “Structures of the Mycobacterial DarR regulator reveal unique modes of second messenger and DNA binding” report crystal structures of different DarR regulators. Some of the structures are in presence of the regulator, cyclic-di-AMP, which the authors claim that this is the first structure of a transcription regulator in presence of this ligand. Similarly, structure in presence of cAMP is also presented. The overall crystallographic study has been done with sufficient rigour. Authors obtained crystals of *M. smegmatis* DarR which diffracted upto 3.56 Å and *M. baixiangningae* DarR which diffracted much better. Authors also solved the structure of DarR from *Rhodococcus* Sp. in presence of c-di-AMP. All the structures show similar mode of binding of c-di-AMP and c-AMP to the proteins, and moreover, the modes of binding of cAMP and c-di-AMP are also similar at the same binding site.

We thank the reviewer for their very helpful comments and critiques. Just to quickly clarify a point, we do not claim in the manuscript that this is the first structure for a transcription regulator bound to c-di-AMP; rather we wrote that this is the first transcription regulator shown to bind c-di-AMP or identified as a c-di-AMP binding protein, which was claimed by Zhang et al. (reference 11).

1). Authors observe that c-di-AMP binds at the interface between two different dimers and propose that this leads to polymer formation. However, no data is presented to show that indeed c-di-AMP leads to aggregation of the proteins. Authors will need to demonstrate this using solution data, say by light scattering in presence of c-di-AMP, or by SEC-MALS studies.

This was not meant to be a key point in the manuscript but an observation of crystal packing. Using a SEC method would require having, minimally, μM concentrations of c-di-AMP in all solutions (including the buffer solutions), which would be prohibitively expensive. We utilized glutaraldehyde crosslinking experiments, which indicate stimulation of higher order oligomers under conditions of c-di-AMP addition (below). However, these experiments are still at high protein concentrations. We do not wish to include and make this a key point of our paper. Also, our data indicate that cAMP may be the central signaling molecule sensed by DarR.

Supplementary Figure 11. Glutaraldehyde crosslinking of DarR \pm c-di-AMP. Left is the MW marker and corresponding MWs for each marker band (The MW marker is from: Bio-Rad, Precision Plus Proteins Dual Color Standards, Batch 64514591 cat # 6110374). First three lanes are *Rhodococcus* DarR at 0, 15 min and 1 hr 0.3% glutaraldehyde. Next three lanes are *Rhodococcus* DarR + 1 mM c-di-AMP at 0, 15 min and 1 hr in 0.3 % glutaraldehyde. The monomer, dimer and higher order bands are indicated at right. The gel is a mini-protean TGX Stain-Free (Bio-Rad) 4-20% precast gel (Cat#456-8093).

Also, on this point, we have included a new Supplementary figure, in which we modeled the polymers onto the dimer-of-dimers DNA-bound DarR. This figure shows that c-di-AMP mediated DarR polymers can form without clash in only one direction when the protein is bound to DNA and that there would be clash in the other direction. Analyses of the polymer without clash however indicates that it would likely not elicit much roadblock function due to the distance/angle of the polymer from the DNA. This is added to the text.

2). The binding affinities of different proteins to c-di-AMP and c-AMP are not very different. This raises doubts over whether c-di-AMP is really the master regulator of these proteins, or if c-AMP can also serve the same with equal efficiency. One wonders what are the intracellular concentrations of cAMP and c-di-AMP, so that specifically only one of them would trigger transcriptional response.

We have now included data on intracellular concentrations of c-di-AMP and cAMP in *M. smegmatis*. Based on these data, the mid- μ M binding affinity of DarR for cAMP suggests that cAMP may be the functionally important second messenger as intracellular concentrations of cAMP in *M. smegmatis* have been estimated to reach as high as 3 mM (reference 61). By contrast, c-di-AMP levels in *M. smegmatis* have been measured in the nM range (reference 62). Hence, even if the K_d of DarR for c-di-AMP is 2.3 μ M, as determined by Zhang et al. (reference 11), DarR may be less responsive to this second messenger. However, it is also important to point out that studies have shown that high target specificity for second messengers can be achieved with local signaling events between specific cyclases and target proteins. These findings indicate that if the second messenger generating enzymes are localized proximal to the target or localized with the target via a direct interaction between the two proteins, the high local concentration of the generated second messenger would permit binding and effects of second messengers to even low affinity binding targets.

3). From Page 12, authors consistently write that DarR binds DNA as dimer of dimers. However, it is seen that two independent dimers bind to DNA. In the larger context, the term “dimer of dimers” is typically used for tetrameric proteins, where two dimers of a protein associate with another dimer of the same protein to form a tetramer.

We thank the reviewer for this comment. The “dimer-of-dimer” terminology has been conventionally used in regard to TetR proteins that bind DNA as two dimers. For example, see Yeo et al., NAR, 2017- “Based on structural analyses, the TetR superfamily transcriptional regulators are divided into two sub-classes depending on their DNA binding mode. One sub-class, including *E. coli* TetR, *S. antibioticus* SimR and *P. aeruginosa* DesT, binds to their cognate DNA as a dimer. The other sub-class, including *S. aureus* QacR, *E. coli* SImA, *S. coelicolor* CprB, *C. glutamicum* CgmR and *Mycobacterium smegmatis* Ms6564, binds as a dimer of dimers.”). However, prior to DarR all previous TFR dimer-of-dimers showed no interactions between dimers. Below is a figure showing examples of the two TFR family DNA binding categories that had been found prior to our studies - that of a dimer and non-interacting dimer-of-dimers. Our studies show a heretofore unseen dimer-of-dimers binding mode where the two TFR proteins directly interact. However, we have provided clarification in the revision on this point.

Examples of 2 TFR DNA binding categories prior to DarR-DNA structures

4). Dimers don't bury 1300 Å² surface from the solvent (Page 6, line 7), the two monomers in a physiological dimer do. Please correct the sentence accordingly.

We thank the reviewer very much for catching this. We have corrected this.

5). There is some worry about the Apo M. smeg DarR data. The CC(1/2) at the cut-off resolution is 0.357, which is lower than the standards that are typically used (although there has been a debate on the inclusion of weak intensities in refinement). This is also reflected in the R_{sym} and R_{merge} values, and during refinement R_{work} and R_{free} values, which are on the higher side. Perhaps it might be good to lower the resolution limit and refine the structure against that resolution data.

Yes, this has been controversial with Karplus/Diederichs suggesting that high-resolution data cutoffs be made to CC1/2 values as low as 0.1. We tried re-refining the structure at a several resolutions, including 3.40 Å, 3.56 Å and 3.75 Å (as there is some weak data in the 3.3-3.4 Å range). While the structures ended up being essentially the same, the refinement with 3.56 Å produced the best results. The lower resolution refinement resulted in poorer statistics, in particular the clashscore.

6). In the Apo- DarR-tris-boron-glycerol structure, the α , β , γ values for trigonal space group are reported as 90, 90, 90. Authors should look into the correct assignment of space group here, or if it is simply typographical mistake.

We thank the reviewer very, very much for catching this typo. Indeed, the space group is P3(2)21 and the gamma angle is thus 120 degrees. We have fixed this in the revision.

7). The PDB validation report states that the space group is P3221, whereas the Table in Supplementary data mentions as P3121.

The correct space group is P3(2)21 and this has been fixed in the table.

8). In Supplementary Figures 1 and 4, the x-axis marking intervals should be identical in all the panels.

As requested, we have replotted these data (now Supplemental Fig. 2 and Supplemental Fig. 7) to place on the same x- and y-axes range.

9). The language of presentation needs improvement. For example, line 12 on Page 5, the sentence is terribly convoluted. Similarly, line 7 on Page 6.

We have rewritten these sentences and hope this has improved the clarity. We also thank the reviewer very much for all their helpful comments.

Reviewer #2 (Remarks to the Author):

In their manuscript Maria Schumacher and colleagues present structures of a TetR-family transcription factor (DarR) with Ligands and supplement this structural part with some biochemical assays to address ligand and DNA binding. DarR has been identified by Zhang et al already in 2012 as the first c-di-AMP binding transcription factor. These authors could show by crosslinking experiments that DarR binds to c-di-AMP and regulates set of genes involved in fatty acid synthesis, a cold shock protein and an MDR transporter). In more detail, c-di-AMP binding to DarR leads to an active repressor (i.e. c-di-AMP induces binding of DarR to its operator site). The manuscript by Schumacher et al. now presents structural data of DarR (in total 6 X-ray structures) of the apo-DarR protein, the DarR-cdiAMP complex and the DarR-dsDNA complex supplemented with Ligand-binding data (fluorescence polarization). The authors furthermore observe a dimer-of-dimer mechanism with dimer-DarR contacting a neighboring (second) DarR-dimer on the DNA site.

The manuscript is clearly written, and the presented structures are described in detail. The authors solved DarR structures in the different states with DarR originating from different organisms, *M. smegmatis* (Msm), *M. baixiangningiae* (Mba), *Rhodococcus* sp (Rsp), which reads a bit unusual (of course sticking to one construct/organism would be more ideal), however, if the constructs failed to crystallize this is a not too unusual approach. To justify the choice of different organisms to be part of one story, the authors provide data showing that the three orthologs behave more or less identical.

In a nutshell, the key findings are

- a) the activator site can bind an unknown ligand that remains to be identified
- b) c-di-AMP and cAMP bind to DarR at the 4helix bundle, however with medium and comparable affinities in the micromolar Kd range
- c) c-di-AMP might bridge two neighboring DarRs
- d) nucleotides slightly increase the affinity to the operator DNA
- e) dimer-of-dimer DarR on dsDNA site shows interaction of facing dimers, however the second one binds to a non-optimal sequence
- f) ligand induced distance between the HTH-sites as a regulatory model for DarR.

We thank the reviewer for their helpful suggestions and critiques.

1). However, I have some points the authors should address – I have to admit it would have been easier if the pages had consecutive line numbers instead of starting with 1 on each page.

We apologize for the line number issue and we thank the reviewer for pointing this out. This has been fixed in the revision.

2). Pg2, Line 8+9: Is there something missing? The sentences read a bit truncated “TFR proteins use dimers to bind DNA”. “the binding mode suggested..”

We have rewritten these sentences to clarify these points. Thank the reviewer for pointing this out.

3). Pg4., line 1: degraded by three families of PDEs - maybe also state that c-di-AMP can be exported by MDRs .

We thank the reviewer for this comment. As suggested, we have added such a sentence to the revision.

4). Pg5. line 14: I know there is an alignment of many DarR homologs, but as the authors claim that *M. smegmatis*, *Rhodococcus* and *M. baixiangningiae* DarR are virtually identical, it would be extremely helpful for the reader to provide the MSA of just these three DarR proteins– or at least group them in Suppl. Fig. 3.

We added a new Supplementary Figure (Supplementary Figure 1) that provides a multiple sequence alignment (MSA) of just the three DarR homologs used in our study.

5). Maybe it is obvious, but the reader might need some info if the other species really possess a c-di-AMP pathway. I guess they all have DACs (?).

We thank the reviewer for this excellent suggestion. We now include a multiple sequence alignment (new Supplementary Figure 6) of the DisA homologs from each of the three species and discuss this in the manuscript. Residues shown to be important for function of the *M. smegmatis* DisA enzyme are also highlighted in this MSA Supplementary Figure (which reveals that these residues are conserved in the *Rhodococcus* and *M. baix.* DisA proteins).

Along these lines, we also included information on *M. smegmatis* adenylyl cyclases (which generate cAMP). *M. smegmatis* harbors eight putative adenylyl cyclases. The *M. smegmatis* MSMEG_3780 is the adenylyl cyclase that has been the best studied; MSMEG_3780 was demonstrated to indeed function as an adenylyl cyclase and to play a role in acid stress in *M. smegmatis*. *Rhodococcus* and *M. baix* both encode MSMEG_3780 homologs. This information has been added to the text and as a new Supplementary Figure (Supplementary Figure 11).

6). Pg5: The authors used the DarR operator site determined by Zahng for *M. smegmatis*. Are these identical for Msm and Rsp? I agree that the FP experiments show almost identical binding, but, again, it is not clear if they share the same operator site. The shape of the curves in FP looks different for Msm – but this is due to the different styles and X-axes (either use identical gridline and spacing or omit them completely, please use identical axes). Maybe I am wrong, but the fitted curve looks sigmoidal at lower concentrations (?) – which model did the authors use to fit the data?

Zhang et al. showed data that *Rhodococcus* DarR binds the same operator site and identified an operator in the *darR* promoter. Our findings support that the homologs all bind the operator with similar high affinity. Notably, the multiple sequence alignment shows that residues that make DNA contacts, in particular those that make base contacts, are conserved among DarR homologs. We have replotted the data to place on the same axes.

7). Is anything known about the regulated genes in *Rhodococcus* sp and *M. baixiangningiae* ? Are they identical to Msm and even more important – do they make sense in the context of c-di-AMP?

We could find nothing in detail about gene regulation in these organisms. However, Zhang et al (reference 11) showed that *Rhodococcus* bind the same, conserved DNA consensus motif and similar to *M. smegmatis*, binds to promoter to autoregulate its expression.

8). Pg6, line 9: please provide a superposition of 3VPR (supp Fig.)

We have added this as a new Supplementary Figure (Supplementary Figure 3), as requested.

9). Pg7: the borate complex is interesting, and it seems well coordinated in the effector binding pocket, the omit density is nice (how did the authors calculate the omit density?). It will certainly be interesting to see what the genuine ligand is. On the other hand – can the different distances between the recognition helices be explained by crystal packing?

The mFo-DFc omit electron density was generated in Phenix; First, the ligand was removed and the structure was subjected to 30 cycles of refinement to limit bias prior to calculating the Sigma-A weighted difference map. We have added this information to the figure legend. The conformational changes observed in this structure caused by ligand binding appear to dictate the orientations/positions of the HTH domains, leading to the generation of these crystals. Otherwise, we might expect that if the packing were controlling the conformation, we should be able to obtain crystals of the apo form (without the ligand) and we cannot.

10). Pg 8+9: the authors could solve the MX structure of *Rhodococcus* DarR with c-di-AMP. The density for the adenine moieties at the helix bundles seems well defined, and the fact that the loop regions become ordered upon ligand binding is a good indication for a binding site. To be able to address the ligand coordination a bit more, it would certainly be beneficial to have coordination distances in the figure (or an additional Ligand-plot). Judged from Fig. 2D the c-di-AMP is mainly kept in place by polar interactions to the sugar/phosphate and more hydrophobic environments at the base, however without the typical stacking of the aromatic residues like the W177 in the direct vicinity. Having everything of the c-di-AMP molecule in one figure is obviously tricky, but it might be hard for the reader to imagine “the truncated rest of c-di-AMP”. The crystallographic table does not include it – what are the B-factors for the protein compared to c-di-AMP and waters and how does the fofc-difference density look like if c-di-AMP is built in?

We have added a sentence on the B-factors of the c-di-AMP compared to protein and water as well as for the cAMP, protein and water in the cAMP structure. We apologize for not making clear in the original document that there are no contacts to the phosphates (hence the poor density). If we fit the c-di-AMP (both phosphates) we see negative mFo-DFc density. We preferred to build the model conservatively. We have added a Supplementary Figure (Supplementary Figure 8) showing this density and also highlighting that the phosphates are exposed to the surface. We also included a new Supplementary Figure for both the DarR-c-di-AMP and DarR-cAMP complexes showing distances (both for hydrogen bond interactions and van der Waals interactions). This is Supplementary Figure 9. The figure shows that the c-di-AMP is held in place by multiple contacts from the protein to the base and a few to a sugar hydroxyl (again, no phosphate contacts).

11.) While I do not doubt the location of the ligand binding as the authors also test it with a mutant protein (W177A etc.), I am not really sure about the physiological role of c-di-AMP with a receptor molecule that has such an unusually low binding affinity compared to e.g. other specific receptors (*B.subtilis* DarA and also the transcription factor *S.agalactiae* BusR e.g. have low nanomolar Kds) Taking all the titrations with c-di-AMP (and also the experiments with cAMP) into

account, I am not convinced that c-di-AMP is the genuine ligand but a “similar” ligand to the genuine. Being not a mycobacterium expert – what are the c-di-AMP concentrations in the cells? Do the K_ds physiologically make sense in the context of the c-di-AMP concentration in the cell? In order to function as an activator for the repressor (e.g. facilitate binding to the operator site), it needs to be in the same range.

This was raised by reviewer 1, point 2. We have included information from studies in *M. smegmatis* on the concentrations of these second messengers. We note that the mid- μ M binding affinity of DarR for cAMP suggests that cAMP may be the functionally important second messenger as intracellular concentrations of cAMP in *M. smegmatis* have been measured to be as high as 3 mM (reference 61). By contrast, c-di-AMP levels in *M. smegmatis* appear to be present in the nM range (reference 62). Hence, even if the K_d of DarR for c-di-AMP is 2.3 μ M, as measured by Zhang et al. (reference 11), DarR may be less responsive to this second messenger compared to cAMP. However, it is important to point out that studies have shown that high target specificity for second messengers can be achieved with local signaling events between specific cyclases and target proteins. These findings indicate that if the second messenger generating enzymes are localized proximal to the target or localized with the target via a direct interaction between the two proteins, the high local concentration of the second messenger would permit binding and effects of second messengers to even low affinity binding targets.

12.) I know the c-di-AMP mediated bridging of DarR observed in the crystal-packing sounds tempting and a polymerisation behavior can indeed increase the local concentration of binding sites. However, did the authors see any oligomerisation tendency of DarR in presence of c-di-AMP in vitro to justify their polymerisation hypothesis (e.g. mass photometry, SAXS/DLS, analytical SEC(-MALLS) in presence of c-di-AMP)?

See response to reviewer 1, point 1. This was not meant to be a key point in the manuscript but, as noted by the reviewer, an observation of crystal packing. Using a SEC method would require having μ M – mM concentrations of c-di-AMP in all solutions (including the buffer solutions), to ensure binding saturation during the experiment, which would be prohibitively expensive. We do not have access to mass photometry. And high affinity interactions, nM, are needed for complex formation in the low concentration schemes used for this technique. But we utilized glutaraldehyde crosslinking experiments, which indicate stimulation of higher order oligomers under conditions of c-di-AMP addition (see response to reviewer 1, point 1). However, these experiments are still at high protein concentrations. We do not wish to include and make this a key point of our paper. Also, as our data indicate that cAMP may be the central signaling molecule sensed by DarR.

In addition, on this point, we have included a new Supplementary figure, in which we modeled the polymers onto the dimer-of-dimers DNA-bound DarR. This figure shows that c-di-AMP mediated polymers can form without clash in only one direction and that there would be clash in the other direction. Analyses of the polymer indicate that it would likely not elicit much roadblock function due to the distance/angle of the polymer from the DNA.

13.) The operator DNA site is also comprising just one recognition site if I got it correctly – so a tandem side-by-side DarR has no advantage over a singular one as they compete for the identical site on the same side of the DNA.

We apologize for not making this point clearer. Actually, there are two overlapping sites for DarR binding on the operator and we now include the sequences of the in vivo characterized operators, which show this characteristic. In these sequences, one recognition site has optimal

sequences in each half site for binding the first DarR dimer. The second site (colored gray and cyan in Fig. 7 and Supplementary Figure 16) has one half site and one non consensus half site. Optimization of that second binding site leads to enhanced binding (Fig. 6). The binding of the first DarR dimer would be predicted to enhance binding of the second site as it impacts the structure of the DNA but also generates a conformation that allows the formation of the cross contacts between the two dimers. Previous structures of TetR proteins that bind as dimer-of-dimers revealed no direct contacts between the dimers. Yet cooperativity has been proposed for several of these dimer-of-dimer TetR binders and since the dimers in these structures do not interact, the hypothesis is that cooperativity arises through DNA alterations; i.e. when the first dimer binds it favors widened major grooves, which we also see in the DarR-DNA complex, that makes it easier for the second dimer to bind. But in the case of DarR, we also see direct interactions between dimers, which would favor both dimers binding simultaneously. In addition, these cross interactions would be disrupted upon inducer binding.

14.) Pg10. With respect to the FP-data I would assume that the Kds are in the same range. I cannot really see it in the figures, but from the figure it seems that cAMP is somewhat better coordinated and that Y178 has a stacking interaction (is it also there with c-di-AMP?). Again, a plot with ligand and coordination distances to the involved residues (and/or a ligand plot in the supplement) would be helpful.

Rather than a schematic plot, which would not provide 3D context, we have added a Pymol figure that includes all the contacts and distances in the Supplementary. We thank the reviewer for the suggestion.

15.) Pg11/12: The ligand induced activation of DarR. In fact, cAMP seems to be a more potent activator of the repressor than c-di-AMP in this functional assay (also see comment to the non-saturating FP data). I think the reported Kds for the DNA-binding make sense. Did the authors test a random DNA-sequence of the same length as a control?

We are not clear as to what this experiment would reveal because the facilitation of dimerization by cAMP/c-di-AMP would be expected to enhance DarR binding to even nonspecific DNA.

16.) The crystal structures of the dimer-of-dimers are interesting, and the SEC analysis shows that DarR without DNA is dimeric and that two of these dimers can bind on one operator dsDNA. I have to admit that it is quite uncommon to perform analytical size-exclusion chromatography on a preparative 320mL 26/60 column instead of using e.g. 24mL or smaller analytical column (maybe in combination with static light scattering, because it seems that protein/DNA amount is not an issue), but I think the Rh-derived data are ok. However, the plot Fig 4C has CV (column volumes on Y-axis) and the data presented in the supplement all show elution before 1CV (320mL column), but Fig 4C has data with 2.4 CVs in the LogMw vs CV plot – that is somehow inconsistent as DarR elutes not at 2.35 CV (=more than 650 mL). I guess there is some mix-up with the column volume.

We thank the reviewer for catching this mix up, which has been fixed in the revision.

17.) DarR intersubunit contacts

The DarR-DarR intersubunit contacts make sense and are interesting. The mutant A119E,L120R makes sense and the FP- and SEC data show the effect of disrupting the dimer of dimers. I am not sure how this would look like in more real concentrations (the authors used 150 μ M DarR which is probably orders of magnitude beyond the concentration in the cell), but the FP data also show the

effect at lower concentration. I think suppl. Figures S8 and S7 show identical SEC-runs for the wt DarR – maybe include the mutant in S7 as S8 is mostly redundant. Even though it is only a supplement, I would definitely ask the authors for proper graphs/plots instead of providing plain screenshots taken from the chromatography software that even show (annotated) noise peaks etc.

As requested by the reviewer we have combined redundant runs and have provided the proper plots for these Supplementary images. The result is new Supplementary Figure 13.

18.) There is one major data fitting issue in Fig 5B. The fit clearly does not describe the measured data. The measured data show a “kink” like usually seen for a stoichiometric titration (i.e. the equilibrium is always on the complex side, high affinity). It looks like the red curve (wt) shows two kinks (one at very low DarR concentration, the second one at around 150nM), while the mutant only shows one. Usually, the stoichiometry can be derived from the FP data if it is a high-affinity binding (see above), but the kink is at approx. 200nM and the DNA-concentration used is 1nM? I am not sure what this means. But as long as the fit does not describe the data – it either a wrong binding model is used or something else is strange – making me believe that the Kds are different than reported.

This binding isotherm is for WT DarR binding to the cognate operator, which we have performed multiple times. We have done the experiment many times and, like other protein-DNA binding FP assays, we see variability in the individual points within the runs. But, we have rerun the experiment yet again and provide a new curve, which does not change the Kd.

19.) Pg14: DarR-DNA contacts

If the second DarR only has half the contacts compared to the “first binder”, also reflected in the weak electron density, I assume that the second binding event is possible but not favoured, however stable in SEC conditions supporting that it happens at least at high concentrations. By mutation of the operator site opposite of the first binding DarR to optimise the second binding event the authors try to increase binding (“optimised” sequence). The FP data of the optimised DNA show slightly better binding compared to the genuine sequence, while the T-mutant and G-mutants disrupt binding. There are two things in the figure: the optimal sequence is not indicated by green crosses as stated in the figure legend: they are black in my PDF. And secondly, the T- and G- mutant data do not serve a fit curve as this is no binding curve. The authors should state that this line is just used for a better visualisation, but is not a binding curve.

As for previous TetR proteins that bind as dimer-of-dimers, there is enhanced binding by the second molecule if the first dimer induces a favorable conformation of the DNA for binding. In the case of DarR, there are cross contacts between dimers that favor binding. We have changed the labeling for the figure with the crosses and removed “fit” for the mutants.

20.) The ligand induced change (pendulum movement) in the distance of the HTH-motifs makes sense, but also underlines that cAMP/cdiAMP are not the main inducers but rather so the ligand at the “ligand binding pocket” (*M. baixiangningiae* structure). I guess it did not crystallise and is certainly not easy to get, but the c-di-AMP+DarR+dsDNA structure would maybe also be helpful, but of course this is rather tricky, and I do not ask for it, here.

Indeed, we are performing crystallization trials of this (and the DarR-cAMP-DNA complex). Thus far we have only obtained crystals of the DarR-cAMP-DNA that diffract to around 15 Å. But we went through multiple crystal forms of the *Rhodococcus* and *Mbax* DarR-DNA complexes before finding ones that diffracted well enough for structure solution. So we shall continue

these efforts.

21.) The discussion makes sense, especially with respect to the high cAMP concentrations in the cell that render the cAMP regulation of DarR more likely than being regulated by c-di-AMP. The authors clearly state that the c-di-AMP triggered polymerisation model lacks some more data at the low concentrations present in the cell (I would suggest trying mass photometry).

See response to point 12 and also our response 1 to reviewer 1.

22.) Figures in general:

The figures, especially the layout of graphs would benefit from some corrections. If the authors like the gridlines that is fine (but not necessary), but they should have uniform spacing and X-axes (especially if they show comparable experiments like titrations). I also assume that the “micro” (i.e. the Greek character μ) is available in any plotting/fitting software package – the authors should use of μM instead of uM

As suggested by the reviewer, we have remade the plots to ensure figures showing comparable experiments have uniform spacing and axes (e.g. Supplementary Figure 2 and Supplementary Figure 7). We have also changed uM to μM in plots using such concentration ranges. We want to again, thank the reviewer for his/her very helpful comments and suggestions.

Reviewer #3 (Remarks to the Author):

DarR was identified in *Mycobacterium smegmatis* as a cyclic-di-AMP binding protein, the first receptor found for this bacterial signalling molecule [citation #11, 2013]. Here, the authors describe six crystal structures of DarR, a TetR family regulator, from three bacterial species (*M. smegmatis*, *M. baixiangningiae*, and *Rhodococcus* sp. USK13, a Gram-positive bacterium related to mycobacteria), together with data from DNA- and nucleotide-binding experiments and size-exclusion chromatography.

The key results are:

- (1) The “apo” structures of the Msm and Mba proteins show that DarR forms a dimer, as expected for TetR family members, but the positions of the DNA binding domains of the two proteins differ by $\sim 10\text{\AA}$.
- (2) The “apo” structure of Mba-DarR shows density in the expected inducer-binding pocket and has been interpreted as a complex of tris buffer and glycerol bound to boron.
- (3) Two structures of the Rho-DarR protein crystallized with cyclic-di-AMP and cAMP show density at a site that is different from the expected inducer-binding pocket.
- (4) Cyclic-di-AMP and cAMP enhance the binding affinity of Rho-DarR to DNA containing the operator sequence by 3- and 11-fold, respectively, from a K_d of 11 nM in the absence of nucleotide.
- (5) Crystal structures of Mba- and Rho-DarR bound to DNA show two dimers of DarR binding to each DNA fragment. The structures show a protein-protein contact between one monomer of each dimer.
- (6) Size-exclusion chromatography indicates that DarR is a dimer in the absence of DNA but is a tetramer (dimer-of-dimers) when bound to DNA.
- (7) Rho-DarR containing mutations designed to disrupt the interaction between dimers binds DNA with a 6-fold lower affinity, and size-exclusion chromatography is consistent with a single dimer of the mutated DarR binding to DNA.
- (8) In the DNA co-crystals structures, one DarR dimer binds to the palindromic consensus binding site while the other dimer binds to a pseudo-palindromic sequence that is shifted relative to the

other site by 3 basepairs. DarR binds ~2-fold more tightly to DNA mutated to optimize the second binding site.

Based on the distances between the DNA binding domains in each structure, the authors conclude that the apo Msm-DarR structure is in a DNA-bound conformation, with a 40 Å distance between the helix-turn-helix motifs; the two Rho-DarR structures with bound nucleotides are in similar conformations, with ~43 Å distance between the HTH motifs. In the “apo” Mba-DarR structure (with the tris-glycerol-boron complex fortuitously bound to the canonical TetR inducer-binding pocket), the HTH motifs are ~50 Å apart, compared to the ~40 Å distance in the Mba- and Rho-DarR complexes with DNA. The authors propose that apo-DarR binds to DNA, repressing transcription and that cyclic-di-AMP or cAMP stabilize this binding. Binding of an inducer, which has not been identified, would cause the type of conformational change that is observed in the “apo” Mba-DarR structure, releasing DarR from DNA.

Overall, the structural and biochemical data presented are sound and present an intriguing model for the regulation of DarR. The structures have been determined at resolutions ranging from 1.44 to 3.56 Å, with the apo *M. smegmatis* structure being the least well determined. However, the biological/physiological significance of these findings is unclear for the following reasons:

We appreciate the helpful comments and critiques from the reviewer.

1.) DarR homologs are not found in the pathogenic mycobacteria, including *M. tuberculosis* and *M. leprae*.

Thank the reviewer for the comment. The focus of the work is to understand adenine nucleotide second messenger binding by DarR regulators and the mechanism of operator DNA binding. However, while *M. smegmatis* was originally considered nonpathogenic, human infections from this bacterium have been described (see Wallace et al, (1988); *M. smegmatis* has been shown to cause infections in skin, soft tissue and bone and *Rhodococcus* infections in humans such as endophthalmitis, osteomyelitis and subcutaneous abscesses have been reported.

DarR homologs are also found in *Rhodococcus (Prescottella) equi*, *Mycobacterium fortuitum* and *Mycobacterium abscessus*. *R. equi* is a known horse pathogen has been shown to cause disease in humans. *M. fortuitum* causes skin and bone infections and *M. abscessus* is a virulent, rapidly growing mycobacteria that is a cause of human pulmonary infections. Although traditionally considered an opportunistic pathogen, *M. abscessus* is now considered a true pathogen.

2.) It is unclear if cyclic-di-AMP is the nucleotide bound to the *Rhodococcus* DarR complex. On page 8, lines 21-22, the authors say that the density for the adenines was well resolved, but “the density for parts of the phosphate groups was less clear (Fig. 2B-E).” From the images provided, it appears that the dinucleotide is not cyclic. The source of the c-di-AMP used for crystallization is not given in the methods section.

We have added the source for the c-di-AMP (Sigma Aldrich). As noted in response to reviewer 2, there are no contacts from DarR to the phosphates of the c-di-AMP; that part of the molecule is completely solvent exposed. Hence, it appears to be flexible and we did not build parts of the molecule with poor or absent electron density. We have added Supplementary Figures to show more clearly the solvent exposed nature of the c-di-AMP when bound to DarR, the contacts to the nucleotide and the map around the phosphates (see new Supplementary Figures 8 and 9). We also provide a clearer description of the interactions in the revision.

3.) The binding of two DarR dimers to each DNA seems serendipitous. Examples of naturally occurring sequences that contain the kind of overlapping operator sites that are present in the

DNA sequence used for the crystallization experiments would significantly increase the relevance of this binding mode.

In the Zhang et al. (reference 11) paper the authors were not initially looking for the overlapping sites as they did not know that DarR binds DNA as a dimer-of-dimers and so pointed to the pseudo palindrome in the in vivo operators. However, the darR and Ms5347 operators indeed contain the overlapping sites and the cspA operator is just missing one base for the overlapping sites (and as we note, Zhang et al showed it is bound more weakly consistent with this finding). We have added this information as a new Supplementary Figure 16 and also tried to provide a clearer description in the text.

4.) As the authors acknowledge, it is unclear how the concentrations of nucleotides used in these experiments (2-5 mM during crystallization) relate to those that are likely to be present in mycobacterial cells.

Nucleotide concentrations in the mM range were required for crystallization experiments because the DarR protein concentrations that were used for these crystallizations were in the mM range (1.0-1.5 mM). Hence, to assure a 1:1 stoichiometry we needed at least 1.5 mM nucleotide. But, we note also that in *M. smegmatis*, the intracellular cAMP concentrations have been shown to range into the mM range.

Additional comments and questions:

5.) Some of the references cited appear to be incorrect for what the author's claim they are citing. For example, in the discussion (page 18, lines 15-17), this sentence cites references 61 and 62:

We thank the reviewer very much for pointing this out. We have utilized specific references for c-di-AMP and cAMP levels in *M. smegmatis*.

6.) "The mid- μ M binding affinity of DarR for cAMP suggests this may be a functionally important interaction as intracellular concentrations of cAMP in *S. smegmatis* have been estimated to be as high as 100 μ M."

However, neither reference ever makes the claim of cAMP concentrations in *M. smegmatis*. Reference 61 is specific to *M. bovis* BCG and *M. tuberculosis* cAMP levels, while reference 62 is about cytokinin signaling and does not mention the words "cAMP" or "cyclic" in the paper. All citations should be checked.

See response to comment 1 above. We have added references describing c-di-AMP and cAMP levels in *M. smegmatis*.

7.) Note: "*S. smegmatis*" in the above sentence should be corrected to "*M. smegmatis*".

We thank the reviewer for pointing out the typo, which has been fixed.

8.) Since DarR is not ubiquitous among Mycobacterial species and the bulk of the structures presented in this paper are of Rhodococcus DarR, the title seems a bit misleading.

Three of the six structures are of DarR from mycobacterial species including DNA-bound and induced forms. Our assays also show that the three orthologs behave essentially the same in terms of DNA and second messenger binding with key residues involved in DNA and cyclic adenine nucleotide binding being conserved among the three. Nonetheless, we take the reviewer's point and have changed the title to "Structures of the DarR transcription regulator reveal unique modes of second messenger and DNA binding".

9.) Is there evidence of any cooperative DNA binding by DarR, especially in the presence of c-di-AMP or cAMP? Do any other adenosine compounds bind to DarR, such as AMP or ATP?

Our binding assays did not explicitly assess cooperative binding, and we do not want to speculate on this point. Zhang et al. showed no ATP binding by DarR up to 1 mM concentrations via SPR (reference 11). However, their DNA binding EMSA assays showed some apparent, though weak, stimulation of DarR binding by adding 500 μM ATP. Hence, we utilized a fluorescent ATP molecule (tagged at the same place as the F-cAMP molecule we used) and performed FP. As you can see below, we do see some indication of binding. However, there was no saturation up to 1 mM added DarR. This suggest that ATP can bind, but weakly, which is not surprising given our structures. The literature suggests that cAMP in *M. smegmatis* (mM) concentrations would exceed those of AMP (measured in bacteria to be in the nM to μM range).

10.) Page 8, lines 2-4: Ms5347 is annotated as an MDR pump. If there are not experimental data testing this, it should be referred to as a putative pump.

Done.

11.) Page 8, line 13: Include the K_d that was previously reported so that the reader does not have to look it up.

We have included this in the revision.

12.) Page 9, lines 2-3: How is it known that the loops in the Msm and Mba DarR proteins becomes ordered upon cyclic-di-AMP binding when only the structure of the Rho-DarR protein has been determined with nucleotide bound?

The *Rhodococcus* DarR-DNA structure did not contain c-di-AMP (or cAMP) and the loop regions in this structure, like the Msm and Mba, have high B-factors and poor density in comparison to the rest of the structure.

13.) Page 9, lines 5-21: This paragraph is rather confusing. One adenine of the di-AMP is described as being bound in a pocket and the other as being solvent exposed, but later in the paragraph the solvent-exposed adenosine is described as being inserted into the binding pocket of an adjacent DarR dimer. How can it be both solvent exposed and inserted into the binding pocket of a symmetry-related molecule?

We apologize for the confusion here. In each DarR subunit, only one adenine base of the c-di-AMP is bound in the pocket of a DarR dimer with the other exposed. The exposed base interacts with a symmetry related molecule in the crystals (hence the hypothesis regarding polymerization arising from crystal packing). We have tried to be clearer with this description in the revision.

14.) Is this binding mode possible in the dimer of dimers bound to DNA that is described later? Given the view of the di-adenine shown in Fig. 2 D-E, it does not seem possible for it to be cyclic. The figures should show the contacts between the nucleotide and the symmetry mate as well as the contacts that are shown. What part of the protein could the second phosphate be contacting, if it is still present and the di-nucleotide is still cyclic? It is not clear how far apart this binding site is from the canonical ligand binding site. Please clarify.

This is an excellent point and we thank the reviewer for this comment We have included a new Supplementary Figure showing that the dimer-of-dimer bound DarR can form interactions via c-di-AMP in one direction but there would indeed be clash in the other direction (please see Supplementary Fig. 14).

As noted for reviewer 1, this was not a focus of the work, and just an observation of crystal packing. The new supplementary figure is helpful in clarifying that any such polymer likely would not elicit enhancement in repressor function due to the distance/angle from the DNA. Rather examination of the interacting dimer-of-dimers binding mode shows that this binding mode would be beneficial in roadblock function. We also apologize for not making clear in the original document that there are no contacts to the phosphates from DarR, likely explaining the poor density. If we fit the c-di-AMP (both phosphates) we see negative mFo-DFc density. We built our model very conservatively. We have added a Supplementary Figure showing this (density and also highlighting that the phosphates are exposed to the surface). We also included a new Supplementary figure for both the DarR- c-di-AMP and DarR-cAMP complexes showing distances (both for hydrogen bond interactions and van der Waals interactions). The figure indicates the nucleotide is held in place by the multiple contacts from the protein to the base. We hope that this has cleared up the issue and again we thank the reviewer for pointing out the need for more clarity.

15.) Page 14, lines 9-23, and page 15, lines 1-11: This paragraph is also confusing. The DNA sequence shown on line 13 does not match the sequence shown in Fig. 6B. It appears to be the “optimized” sequence that is shown in Fig. 7B. Since the mutated sequences shown are not fully self-complementary, the manuscript should clarify that two complementary DNA oligos are annealed together to form a fully base paired duplex (like what is shown in Fig. 7B for the optimized sequence), if that is what was done. Adding the duplex sequences of the mutants to the figure would be helpful.

We have added clarifying text on DarR operator palindrome in the introduction and expanded the discussion on the in vivo operators. We note, all the operator DNA oligonucleotides used were double stranded, composed of two complementary sequences. We have added this information. We have included all the mutant and in vivo operator sequences in a new Supplementary Figure.

16.) Fig. 6 A and B should more clearly show which DarR molecules make up the dimer, since the dimerization domains are not visible. As labeled, Fig. 6A suggests that the “cross contacts” are between the green and magenta monomers, but aren’t those the monomers that make up one

dimer? Shouldn't there also be a symmetric set of cross contacts? The density shown in Fig. 7A would seem better to include in a figure with panels 6A, B and C.

We apologize for the lack of clarity. We show in Fig 6A only the DNA-binding domains as the inclusion of the C-terminal domains occluded many of the contacts. As in Figs 4 and 5 the green/magenta subunits and the slate/grey pairs each make dimers. We have added labels to indicate this in figure 6A. The cross contacts are between the grey and green subunits, which are shown in this figure but highlighted in Fig. 5. We tried to combine the electron density figures with Fig 6. However, the figure was quite crowded and illegible. So we have kept it as Figure 7 but have added clarifying figure legends and information in the text.

17.) Fig. 8 is not as informative as it could be. Where are the ligand binding sites relative to the structures shown? Adding a DynDom analysis of the movement would be informative.

We have remade the figure and now show the ligands as spheres so that they are easy to see. We thank the reviewer for this excellent suggestion and the excellent suggestion of using DynDom. DynDom showed a rotation of 19.7 degrees on an axis centered on the ligand binding site. The result is a large shift of the ends of the helices 6 and 7 and, in particular, the adjacent DNA binding domain outward upon ligand/inducer binding.

REVIEWER COMMENTS

Reviewer #1 (Remarks to the Author):

All the changes suggested have been made and the questions raised have been answered.

One point remains, that about the affinity with cAMP vs c-di-AMP. It is clear from the response of the authors that primarily these proteins serve as cAMP regulated transcription factors, notwithstanding the local concentration of c-di-AMP written by the authors.

Reviewer #2 (Remarks to the Author):

The revised manuscript by Maria Schumacher shows a substantial improvement compared to the initial submission and lots of the reviewer comments have been taken into account.

I have some points the authors should check again:

line 121/122 "These structures reveal the same dimer": I guess the authors mean that the assembly (M.smegmatis DarR, 2 mol/ASU) can be generated by symop from the M.baixiangningiae structure?

line 188ff: the B-factors of c-di-AMP/the Pi even more suggest that c-di-AMP is likely not as coordinated as cAMP is?

line 292ff: the polymer discussion sounds more reasonable now.

line 443ff: I appreciate and fully support the discussion that cAMP (and not c-di-AMP) is the functional relevant molecule in this context. The "local concentration" part might be correct for some selected messengers, but I personally favour the first explanation. If cAMP is approx 3mM in a small bacterium - then it sounds quite unlikely that local c-di-AMP concentrations rise above the Kd in a background of 3mM cAMP (diffusion is fast for small molecules).

It would be interesting to see a competition experiment: in a preformed complex of fluorescent-cAMP+DarR - > what happens if one adds (non-labeled) c-di-AMP (at different concentrations) and vice versa?

Misc:

- Suppl. fig 1 is now included as suggested and nicely supports the high homology,
- the DAC MSA is included (ok),
- superposition S3 (ok),
- pt12 (oligomers) I like the crosslink experiment - and I think the authors are correct in not overinterpreting the crosslinks at very high concentrations to render this a "key point" of the manuscript.

- my point 18 (regarding figure 5B):

Maybe I somehow did not make it clear in my first review - and this is not criticism on the results of the authors. I do not doubt that the authors repeated the experiment and that a fit yields a Kd - but looking at figure 5B (blue curve) the authors will certainly agree that the fit curve (line) does not at all describe the measured data (!). One can use any algorithm to fit any data and the software always gives a Kd, the question is: does the fit describe the data and is it the correct model. In this case - just to clarify what I meant - and sorry for the long explanation:

This is something that is e.g. also observed for SSB proteins binding to short ssDNA oligos (way before 1990 I think):

Assuming you have A (the protein) binding to B (the fluorescent DNA) yielding AB (which is obviously the case). The equilibrium is $A+B \rightleftharpoons AB$ and the equilibrium constant is $K_d = \frac{AB}{(A) \cdot (B)}$ with (A) and (B) being the concentrations _in equilibrium_ (i.e. $(A) = A_0 - (AB)$). If a ligand is a really high-affinity binder with a very low K_d , then any addition of A (protein) to B will lead to a situation with 100% of the added A bound to B and no free concentration of A is present, leading to a straight line in the "addition phase" of A until the saturation is reached (all B then is occupied) = the stoichiometry point. From this point onwards, any addition of A will not change the signal anymore, as all B is already in complex. From here on $(B) = 0$. This is usually referred to as a "stoichiometric titration". (two straight lines, a kink at the stoichiometry point).

Having no free A in the phase before reaching the stoichiometry means - or $B = 0$: not defined in the K_d equation above (division by zero). In the end fitting algorithms can fit any K_d into these data - the K_d is determined mostly from the shape of the "curvature" which is not existing (its a straight line here). And the saturation is also not well described in 5B (blue curve). Thus a K_d determined by this experiment is unreliable, and if at all just an upper limit of the real K_d . The solution would be to lower the concentrations in the assay - typically this ends up regimes where noise takes over and real detection does not work.

In the end: it doesn't change anything. Its a high affinity binding event and the data clearly show this. However, in this case of a stoichiometric titration, the K_d is not reliable and - if at all - just a upper limit (i.e. the real K_d is probably much lower than the determined one).

To sum up - I think the manuscript has improved a lot and I congratulate the authors to the crystal structures of DarR in different states.

My personal opinion based on the presented data is, that the more realistic ligand of DarR is cAMP rather than cdiamp.

Reviewer #3 (Remarks to the Author):

The authors have largely addressed my concerns, but I have a few final comments:

In Figure 8, the change in color of the residues 108-132 clarifies the conformational change, upon inducer binding, but describing this region as the "hinge point" is not very accurate. Adding the axis of rotation to the figure would clarify the actual hinge point. Also, in this version of the figure, the DNA is not very visible.

The manuscript could use additional proofreading to catch mistakes.

RESPONSE TO REVIEWERS: REVISION 1

Reviewer #1 (Remarks to the Author):

All the changes suggested have been made and the questions raised have been answered. One point remains, that about the affinity with cAMP vs c-di-AMP. It is clear from the response of the authors that primarily these proteins serve as cAMP regulated transcription factors, notwithstanding the local concentration of c-di-AMP written by the authors.

We thank the reviewer again for all their helpful comments and suggestions. We agree with the reviewer that DarR is most likely a cAMP responsive factor and have revised the document further to reflect this point (reviewer 2 indicates this as well).

Reviewer #2 (Remarks to the Author):

1. The revised manuscript by Maria Schumacher shows a substantial improvement compared to the initial submission and lots of the reviewer comments have been taken into account. I have some points the authors should check again: We wish to thank the reviewer very much for the helpful comments and suggestions.

2. line 121/122 "These structures reveal the same dimer": I guess the authors mean that the assembly (M.smegmatis DarR, 2 mol/ASU) can be generated by symop from the M.baixiangningiae structure? We thank the reviewer for pointing this out. Indeed, there is one M. baixiangningiae DarR subunit in the crystal structure with the dimer generated by crystallographic symmetry. This information is now clarified in the text.

3. line 188ff: the B-factors of c-di-AMP/the Pi even more suggest that c-di-AMP is likely not as coordinated as cAMP is? We have revised the manuscript, including the Abstract and Discussion, to reflect this. We agree with the reviewer on this point.

4. line 292ff: the polymer discussion sounds more reasonable now. We agree.

5. line 443ff: I appreciate and fully support the discussion that cAMP (and not c-di-AMP) is the functional relevant molecule in this context. The "local concentration" part might be correct for some selected messengers, but I personally favour the first explanation. If cAMP is approx 3mM in a small bacterium - then it sounds quite unlikely that local c-di-AMP concentrations rise above the Kd in a background of 3mM cAMP (diffusion is fast for small molecules).

It would be interesting to see a competition experiment: in a preformed complex of fluorescent-cAMP+DarR -> what happens if one adds (non-labeled) c-di-AMP (at different concentrations) and vice versa? We have attempted competition experiments- see right. cAMP seems to compete better. But, the assay, which uses relatively high protein concentrations (for saturation) led to aggregation with increasing added cold ligand, under the assay conditions. This assay is unfortunately sensitive to aggregation. We do not wish to include these data due to the technical problems. Also, studies have indicated the intracellular concentration of cAMP are many thousand to million fold higher than c-di-AMP (competition seeming very unlikely). In the revision we indicate cAMP as the physiological effector in agreement with the reviewer.

Addition of cAMP to DarR-F-c-di-AMP
Addition of c-di-AMP to DarR-F-cAMP
F-c-di-AMP or F-cAMP were titrated to saturation binding with *Rhodococcus* DarR and then non-F cAMP (red) or c-di-AMP (blue) were added. Buffer: 25 mM HEPES pH 7.5, 150 mM NaCl, 5% (v/v) glycerol further addition of either cold cAMP or c-di-AMP resulted in aggregation, beyond points shown. The lines between points are not curve fits.

Misc:

- Suppl. fig 1 is now included as suggested and nicely supports the high homology

We thank the reviewer again for this suggestion

- the DAC MSA is included (ok), **ok**

- superposition S3 (ok), **ok**

- pt12 (oligomers) I like the crosslink experiment - and I think the authors are correct in not overinterpreting the crosslinks at very high concentrations to render this a "key point" of the manuscript. **We agree with the above and thank the reviewer for these suggestions and comments.**

6. my point 18 (regarding figure 5B):

Maybe I somehow did not make it clear in my first review - and this is not critics on the results of the authors. I do not doubt that the authors repeated the experiment and that a fit yields a K_d - but looking at figure 5B (blue curve) the authors will certainly agree that the fit curve (line) does not at all describe the measured data (!). One can use any algorithm to fit any data and the software always gives a K_d , the question is: does the fit describe the data and is it the correct model. In this case - just to clarify what i meant - and sorry for the long explanation:

This is something that is e.g. also observed for SSB proteins binding to short ssDNA oligos (way before 1990 I think):

Assuming you have A (the protein) binding to B (the fluorescent DNA) yielding AB (which is obviously the case). The equilibrium is $A+B \leftrightarrow AB$ and the equilibrium constant is $K_d = \frac{(AB)}{((A)*(B))}$ with (A) and (B) being the concentrations _in equilibrium_ (i.e. $(A) = A_0 - (AB)$). If a ligand is a really high-affinity binder with a very low K_d , then any addition of A (protein) to B will lead to a situation with 100% of the added A bound to B and no free concentration of A is present, leading to a straight line in the "addition phase" of A until the saturation is reached (all B then is occupied) = the stoichiometry point. From this point onwards, any addition of A will not change the signal anymore, as all B is already in complex. From here on $(B) = 0$. This is usually referred to as a "stoichiometric titration". (two straight lines, a kink at the stoichiometry point).

Having no free A in the phase before reaching the stoichiometry means - or $B = 0$: not defined in the K_d equation above (division by zero). In the end fitting algorithms can fit any K_d into these data - the K_d is determined mostly from the shape of the "curvature" which is not existing (its a straight line here). And the saturation is also not well described in 5B (blue curve). Thus a K_d determined by this experiment is unreliable, and if at all just an upper limit of the real K_d . The solution would be to lower the concentrations in the assay - typically this ends up regimes where noise takes over and real detection does not work.

In the end: it doesn't change anything. Its a high affinity binding event and the data clearly show this. However, in this case of a stoichiometric titration, the K_d is not reliable and - if at all- just a upper limit (i.e. the real K_d is probably much lower than the determined one). **We have included one of the replicate isotherms with a better curve fit. We and others have done stoichiometric binding experiments via FP, and for these experiments, one uses the probe at 10 fold (or more) above the K_d (to ensure the binding becomes stoichiometric; see for examples, Hoffmann et al. (2005) *J Bacteriol*, 187: 5008-5012 and Tschowri et al. *Cell*, 158: 1136-1147). We should clarify that in this experiment we used only 1 nM F-DNA probe. To ensure clarity on this point, we have added that all DNA binding FP experiments were done with 1 nM F-DNA probe in the text.**

To sum up - I think the manuscript has improved a lot and I congratulate the authors to the crystal structures of DarR in different states. My personal opinion based on the presented data is, that the more realistic ligand of DarR is cAMP rather than cdiamp. **Given all the data and reported**

concentrations in cells, we agree with that assessment and have now edited the manuscript to reflect that our data support cAMP as the likely physiological. We thank the reviewer again for all their great input!

Reviewer #3 (Remarks to the Author):

The authors have largely addressed my concerns, but I have a few final comments: **We really appreciate the reviewer's helpful comments and suggestions.**

In Figure 8, the change in color of the residues 108-132 clarifies the conformational change, upon inducer binding, but describing this region as the "hinge point" is not very accurate. Adding the axis of rotation to the figure would clarify the actual hinge point. Also, in this version of the figure, the DNA is not very visible. **We thank the reviewer for this suggestion. We have remade the figure to make the DNA more visible in Fig. 8A and added a Fig. 8B to show the location of the axis of rotation specifically output from Dyndom as an arrow.**

The manuscript could use additional proofreading to catch mistakes.

We have tried to catch remaining typos and other errors.